# Improving the Reconstruction of Disentangled Representation Learners via Multi-Stage Modelling

## Abstract

Current autoencoder-based disentangled representation learning methods achieve disentanglement by penalizing the (aggregate) posterior to encourage statistical independence of the latent factors. This approach introduces a trade-off between disentangled representation learning and reconstruction quality since the model does not have enough capacity to learn correlated latent variables that capture detail information present in most image data. To overcome this trade-off, we present a novel multi-stage modelling approach where the disentangled factors are first learned using a preexisting disentangled representation learning method (such as $\beta$-TCVAE); then, the low-quality reconstruction is improved with another deep generative model that is trained to model the missing correlated latent variables, adding detail information while maintaining conditioning on the previously learned disentangled factors. Taken together, our multi-stage modelling approach results in single, coherent probabilistic model that is theoretically justified by the principal of D-separation and can be realized with a variety of model classes including likelihood-based models such as variational autoencoders, implicit models such as generative adversarial networks, and tractable models like normalizing flows or mixtures of Gaussians. We demonstrate that our multi-stage model has much higher reconstruction quality than current state-of-the-art methods with equivalent disentanglement performance across multiple standard benchmarks.

## 1 Introduction

Deep generative models (DGMs) such as variational autoencoders (VAEs) (Kingma & Welling, 2014; Rezende et al., 2014) and generative adversarial networks (GANs) (Goodfellow et al., 2014) have recently enjoyed great success at modeling high dimensional data such as natural images. As the name suggests, DGMs leverage deep learning to model a data generating process. The underlying assumption is that the high dimensional observations $X \in \mathbb{R}^D$ can be meaningfully described by a small set of latent factors $H \in \mathbb{R}^K$, where $K < D$. More precisely, the observation $(X = x)$ is assumed to be generated by first sampling a set of low dimensional factors $h$ from a simple prior distribution $p(H)$ and then sampling $x \sim p_\theta(X|h)$. DGMs realize $p_\theta$ through a deep neural network also known as the decoder or the generative network. VAE-based DGMs use another deep neural network (called the encoder or the inference network) to parameterize an approximate posterior $q_\phi(H|x)$. Learning the variational posterior parameters is done by maximizing an evidence lower bound (ELBO) to the log-marginal likelihood of the data under the model $p_\theta(X)$.

Learning *disentangled* factors $h \sim q_\phi(H|x)$ that are semantically meaningful representations of the observation $x$ is highly desirable because such interpretable representations can arguably be advantageous for a variety of downstream tasks (Locatello et al., 2018), including classification, detection, reinforcement learning, transfer learning and image synthesis from textual descriptions (Bengio et al., 2013; LeCun et al., 2015; Lake et al., 2017; van Steenkiste et al., 2019; Reed et al., 2016; Zhang et al., 2016). While a formal definition of disentangled representation (DR) remains elusive, it is understood to mean that by manipulating only one of the factors while holding the rest constant, only one semantically meaningful aspect of the observation (e.g. the pose of an object in an image) changes. Prior work in unsupervised DR learning focuses on the objective of learning statistically independent latent factors as means for obtaining DR. The underlying assumption is

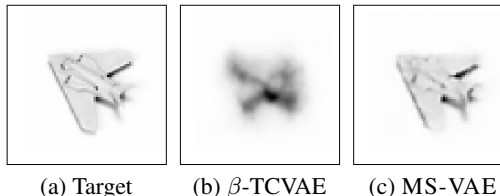

(a) Target     (b) $\beta$-TCVAE     (c) MS-VAE

Figure 1: Image reconstruction using $\beta$-TCVAE (Figure 1b) and MS-VAE (Figure 1c). MS-VAE is able to take the blurry output of the underlying $\beta$-TCVAE model and learn to render a much better approximation of the target while maintaining the pose of the original image (Figure 1a).

that the latent variables $H$ can be partitioned into independent components $C$ (i.e. the disentangled factors) and correlated components $Z$. An observation $(X = x)$ is assumed to be generated by sampling the low dimensional factors $h = (c, z)$ from $p(H)$ and then sampling $x \sim p_\theta(X|c, z)$ (Figure 2a). A series of works starting from (Higgins et al., 2017) enforce statistical independence of the latent factors $H$ via regularization, up-weighting certain terms in the ELBO which penalize the (aggregate) posterior to be factorized over all or some of the latent dimensions (Kumar et al., 2017; Kim & Mnih, 2018; Chen et al., 2018) (see Section 2 for details).

While the aforementioned models show promising results, they suffer from a trade-off between DR learning and reconstruction quality. If the latent space is heavily regularized – not allowing enough capacity for the correlated variables – then the reconstruction quality will be diminished, signaling that the learned representation is sub-optimal. As the correlated variables are functionally ignored with high levels of regularization, an observation $(X = x)$ can be thought to be generated by only sampling independent latent factors $c$ from $p(C)$ and then sampling $x \sim p_\theta(X|c)$ (Figure 2b). This is a departure from the original data generating hypothesis that $x$ is sampled from a distribution dependent on both the independent and correlated latent variables. On the other hand, if the correlated variables are not well-constrained, the model can use them to achieve a high quality reconstruction while ignoring the independent variables (the disentangled latent factors). This phenomena is referred to as the "shortcut problem" and has been discussed in previous works (Szabó et al., 2018; Lezama, 2018). Overcoming the aforementioned trade-off is essential for using these models in real world applications such as realistic, controlled image synthesis (Lee et al., 2020; Lezama, 2018).

In this paper, we propose a new graphical model for DR learning (Figure 2c) that allows for learning disentangled factors while also correctly realizing the data generating hypothesis that an observation is generated from independent and correlated factors. Importantly, the graphical model in Figure 2c is D-separated, meaning that any changes in the correlated latent variables $Z$ will not influence the independent latent variables $C$. Generating an observation $(X = x)$ from this model can then be done by sampling the independent factors $c$ from $p(C)$, sampling a low-quality reconstruction $y \sim p_\theta(Y|c)$, sampling the correlated factors $z$ from $p(Z)$, and then finally sampling $x \sim p_\theta(X|z, Y)$. The final reconstruction $x$ depends both on $z$ and $c$, however, any regularization needed to extract the independent factors $c$ no longer diminishes the model capacity for the correlated factors $z$.

To realize our proposed graphical model, we introduce **MS-VAE**, a multi-stage DGM that is implemented as follows: first, the disentangled representation $C$ is learned using an existing DR learning method such as $\beta$-TCVAE (Chen et al., 2018). Since the learned factors $C$ are regularized to be statistically independent – not allowing enough capacity for correlated factors – the final reconstruction $Y$ will have diminished reconstruction quality. Then, we train another DGM to improve the low-quality reconstruction $Y$ by learning the missing correlated factors $Z$. This is achieved during training by inputting the reconstruction $Y$ into the decoder of the second DGM and then modulating the hidden activation of each layer as function of latent factor $Z$ (using Feature-wise Linear Modulation (Perez et al., 2018)). Through this training paradigm, MS-VAE is able to preserve conditioning on the disentangled factors while dramatically improving the reconstruction quality. A schematic of MS-VAE is shown in Figure 2d and example images from each stage are shown in Figure 1. The reconstruction from $\beta$-TCVAE (1b) is improved by MS-VAE (1c) to better approximate the target (1a) while maintaining conditioning on the disentangled factors, e.g. azimuth.

To summarize our contributions:

- We propose a new graphical model for DR learning (Figure 2c) that, due to the D-separation between its independent and correlated latent variables, alleviates the disentanglement vs. reconstruction trade-off.

- We introduce MS-VAE, a multi-stage DGM that implements our proposed graphical model, achieving state-of-the-art reconstruction quality while maintaining the same level of disentanglement as preexisting methods. MS-VAE can be implemented using a variety of DGMs such as VAEs, GANs, or FLOW-based models and does not depend on hand-crafted loss functions or additional hyperparameters (unlike current state-of-the-art (Lezama, 2018))

- Following (Locatello et al., 2018), we test our framework with a wide-range of qualitative and quantitative tests to demonstrate the efficacy of the framework. We provide all code used for the experiments.

## 2 BACKGROUND

VAEs, first introduced in (Kingma & Welling, 2014), represent a class of likelihood-based, DGMs comprised of an encoder and a decoder that are trained using amortized variational inference to maximize the expected ELBO under the data distribution,

$$\int p(x) \log p_\theta(x) dx \geq \mathbb{E}_{p(x)} \left[ \mathbb{E}_{q_\phi(h|x)}[\log p_\theta(x|h)] - \text{KL}[q_\phi(h|x) \| p(h)] \right] \tag{1}$$

The first term is the expected reconstruction loss under the variational posterior $q_\phi$. The second term – the Kullback–Leibler (KL) divergence – underpins most of the recent DR learning methods. The KL term regularizes the variational posterior to be similar (in expectation) to a simple spherical Gaussian prior over $H$. Higgins et al. (2017) attempts to factorize the variational aggregate posterior by up-weighting this KL-divergence term with a Lagrange multiplier $\beta$. Follow-up works decompose the KL-term in equation 1 and up-weight only the KL-divergence between the aggregate posterior $q_{\phi_H}(H) = \int_X p(x) q_\phi(H|x) dx$ and the factorized aggregate posterior (Kim & Mnih, 2018; Chen et al., 2018; Jeong & Song, 2019). Kumar et al. (2017) simply uses a factorized prior.

These models introduce a trade-off between unsupervised DR learning and reconstruction quality. When the latent variables are heavily regularized to be independent, an observation ($X = x$) can be thought to be generated by sampling independent latent factors $c$ from $p(C)$ and then sampling $x \sim p_\theta(X|c)$ (Figure 2b). This leads to low-quality reconstructions as the correlated factors are not utilized. Lezama (2018) attempts to tackle this issue with a teacher-student learning technique. First, the teacher model is trained to extract disentangled factors at the cost of reconstruction quality (similar to the first stage of our model). Then, a student autoencoder (AE) model with a larger latent dimension size is trained with additional losses that force the Jacobian of the final reconstruction with respect to the disentangled factors to remain the same (a.k.a Jacobian supervision). While this model achieves promising results on MNIST (Deng, 2012), it suffers from a few drawbacks including having to relearn the disentangled factors in the second stage and having no theoretical guarantees that changes in the correlated factors $Z$ will not affect the independent factors $C$ in the student model. Other works have also been proposed to overcome the trade-off using adversarial training (Mathieu et al., 2016; Lample et al., 2017; Perarnau et al., 2016; Szabó et al., 2018; Hu et al., 2018).

## 3 MS-VAE

### 3.1 MS-VAE GRAPHICAL MODEL

**Lower Bound.** To alleviate the trade-off between reconstruction and disentanglement, we propose a new graphical model for DR learning (Figure 2c). To understand this model, let us assume that the data come as pairs of images $\{y_i, x_i\}_{i=1}^N$. Let the $x$'s be the ground truth images (generated using both the independent factors $c$ and correlated factors $z$) and let the $y$'s be the approximations of those images generated using only $c$. Given this paired data, the graphical model depicted in Figure 2c represents a single coherent model of the data that defines a joint distribution over $Y$ and $X$. The log-marginal likelihood of the observations under this model can be lower-bounded as follows,

$$\log p_{\{\phi,\theta,\theta_Z,\Theta\}}(x,y) \geq \tag{2}$$
$$\underbrace{- \text{KL}[q_\phi(c|y)\|p(c)] + \mathbb{E}_{q_\phi}[\log p_\theta(y|c)]}_{\text{(a) } C \rightarrow Y} \underbrace{- \text{KL}[q_{\theta_Z}(z|x,y)\|p(z)] + \mathbb{E}_{q_{\theta_Z}}[\log p_\Theta(x|z,y)]}_{\text{(b) } (Y,Z) \rightarrow X}.$$

In practice, we do not have access to $Y$. However, by learning the disentangled factors $C$ from $X$, we can generate $Y$. Therefore, we first use state-of-the-art DR learning method to learn the sub-graph $C \rightarrow Y$ by maximizing term (a) in Equation 2 to produce $Y$. Then, we train another DGM to learn the sub-graph $(Y, Z) \rightarrow X$ by maximizing term (b) in Equation 2. This two-stage training procedure approximates the graphical model from Figure 2c.

**D-separation.**    Now that we introduced and lower-bounded the graphical model from Figure 2c, it is important to understand the theoretical motivation for realizing this model. Recall that our goal is to take an existing method for learning the disentangled factors C and learn additional residual information in Z without changing the conditioning on C. In a standard VAE graphical model (Figure 2a), this could occur because there is no guarantee that $p(Z,C|X) = p(Z|X)p(C|X)$. By introducing an observed node Y (i.e. D-Separation) in our graphical model, we ensure that changes in Z will not influence C. This is because the joint distribution $p(Z, C|X, Y)$ can now be broken down as $p(C|Y)p(Z|X, Y)$. In practice, realizing the graphical model in Figure 2c using a deep neural network requires care as we want to ensure that the conditioning on Y is stronger than that of Z so that Z only captures the residual correlated information. Also, simply training a single neural network end-to-end to approximate this graphical model is infeasible as Y will no longer act as an observed node, leading to the entanglement of C and Z. We demonstrate this phenomenon by performing an ablation study of MSVAE (Appendix L).

### 3.2 MS-VAE IMPLEMENTATION

In the following sections, we describe how our MS-VAE implements the graphical model in Figure 2c using a multi-stage modelling approach.

**Stage one: Learning a Disentangled Representation.**    In stage one, we train a $\beta$-TCVAE (Chen et al., 2018) to learn the independent latent factors $C$. As $\beta$-TCVAE heavily regularizes the latent space to be statistically independent – not allowing enough capacity for the correlated factors – the final reconstruction $Y$ will be low-quality. Stage one can be realized using any DR learning method, however, we chose $\beta$-TCVAE in our work because it has been shown to perform well across all standard benchmarks and its total correlation penalty is simpler to compute in comparison to FactorVAE's (Kim & Mnih, 2018). We use the standard convolution and transposed-convolutions-based realization of $\beta$-TCVAE as provided in the *disentanglement_lib* (Locatello et al., 2018) package.

**Stage Two: Improving the Reconstruction.**    In stage two, we learn the correlated factors $Z$ with another DGM and then use them to improve the reconstruction $Y$ while maintaining the conditioning on the independent factors $C$. To this end, we train a conditional VAE that models the data $X$ given $Y$ and $Z$. The encoder network $R_{\theta_Z}$ learns the posterior distribution over $Z$ as a function of the residual of $X$ and $Y$. The observation model $G_\Theta$ then reconstructs $X$ given $Y$ and $Z$. Incorporating $Z$ into $G_\Theta$ requires care as simply inputting the concatenation of $Y$ and $Z$ into the network at the start of training will result in a non-linear entanglement between $Y$ and $Z$. Once entangled this way, $G_\Theta$ will fail to condition on $Y$ sufficiently and use the entangled representation as a whole towards the reconstruction of $X$. To overcome this, we induce an architectural bias with a Feature-wise Linear Modulation (FiLM) technique (Perez et al., 2017) prevalent in style transfer literature (Huang & Belongie, 2017), introducing $Z$ only through the adaptive instance normalization (AdaIN) layers of $G_\Theta$. By inputting only $Y$ into the decoder and then incorporating $Z$ through AdaIN at each layer, the network is able to use $Z$ later in the generative process to model the residual information (as explained in (Dumoulin et al., 2018)). This allows the network to better utilize the information in $Y$. We demonstrate the importance of AdaIN by performing an ablation study of MSVAE (Appendix L).

**FiLM and AdaIN.**    Recently, FiLM have been proposed as a general-purpose conditioning technique for deep neural networks (Perez et al., 2018). A FiLM layer learns an affine transformation of the intermediate statistics of a deep neural network, conditioned on an external input. FiLM has found great success in style transfer literature where the goal is to render content from a source domain in the style of a target domain. One such method, Instance Normalization (Ulyanov et al., 2016), incorporates the style and the content information through the normalization layers in a feed-forward network,

learning style-specific affine parameters from the data that are used to shift and scale the feature statistics (per instance) extracted from the source image across spatial dimensions. Huang & Belongie (2017) extended this method to address arbitrary styles by taking the affine parameters to be the feature statistics extracted from the style domain, arriving at the $AdaIN(y, x) = \sigma(x)\left(\frac{y-\mu(y)}{\sigma(y)}\right) + \mu(x)$ which is applied for each normalization layer. Here, $y$ are the incoming activations for a normalization layer in the network that is receiving an image from the source domain and $x$ are the style features computed from the target domain. The features statistics for $y$ (as well as for for $x$) are computed as follows: $\mu(y) = \frac{1}{HW}\sum_{h,w} a_{nhwc}$ and $\sigma(y) = \sqrt{\frac{1}{HW}\sum_{h,w}(x_{nhwc} - \mu(y))^2 + \epsilon}$, where $h, w, c$ stand for the height, width and channel, respectively. In MS-VAE, the equivalent of style features is the representation of the residual information between $X$ and $Y$ captured in the correlated variables $Z$. The goal is similar to style transfer: incorporate the information stored in $Z$ while maintaining the semantic content stored in $Y$. Therefore, we slightly modify the AdaIN formulation as follows: $AdaIN(y, z) = \gamma(z)\left(\frac{y-\mu(y)}{\sigma(y)}\right) + \beta(z)$. where $\mu(y)$ and $\sigma(y)$ are the same as before and $\gamma(z)$ and $\beta(z)$ are learned functions of $z$ parameterized by a fully-connected neural network.

**MS-GAN and MS-FLOW.** While the above section only described the VAE-based implementation of our method, we can also realize these techniques using GANs or FLOW-based models. We defer these implementations to Appendix E and D.

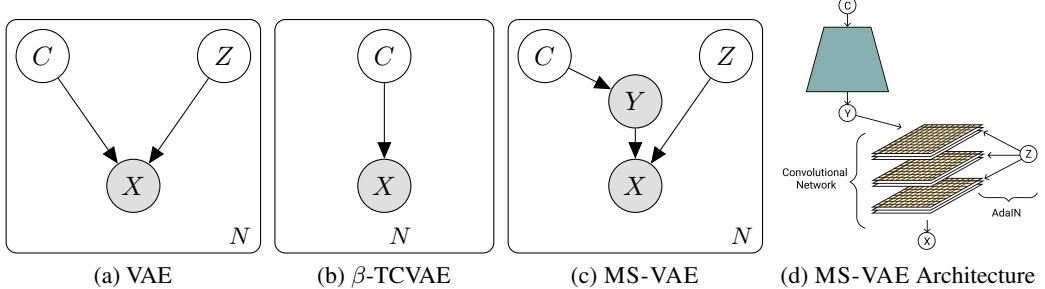

|     |     |     |     |
| --- | --- | --- | --- |
| (a) VAE | (b) $\beta$-TCVAE | (c) MS-VAE | (d) MS-VAE Architecture |

Figure 2: (a) Graphical model of a standard VAE where $C$ and $Z$ are not independent conditioned on $X$. (b) Graphical model of $\beta$-TCVAE where the reconstruction only depends on the independent latent factors $C$. (c) MS-VAE graphical model where $C$ and $Z$ are independent conditioned on $Y$. (d) Schematic for MS-VAE when implemented as a convolutional architecture. Both $Y$ and $X$ are the reconstructions of the same image.

**Deeper MS-VAE.** In this work, we focus on generative models with only two levels of hierarchy and two sets of latent factors $C$ and $Z$. However, MS-VAE can be easily extended to graphical models with deeper or more structured hierarchies. We present a concrete example of this extension of MS-VAE in appendix C where we use MS-VAE to model a simple 2D pendulum.

## 4 EXPERIMENTS

**Baseline Models** Locatello et al. (2018) demonstrated that most state-of-the-art DR learning methods are able to effectively learn a factorized posterior distribution (i.e. extract independent latent variables $C$). Therefore, we chose $\beta$-TCVAE as the baseline DR learning method for our paper. For fair comparison with our work, we use $\beta$-TCVAE to implement the sub-graph $C \rightarrow Y$ in MS-VAE. For our main experiments, we thoroughly evaluate three models: (i) MS-VAE with latent dimensions $C$ for the $C \rightarrow Y$ network and $Z$ for the $(Y, Z) \rightarrow X$ network, (ii) $\beta$-TCVAE with latent dimension size $|C|$, and (iii) $\beta$-TCVAE-L with latent dimension size $|C + Z|$. In line with previous work (Locatello et al., 2018), we use a 10-dimensional $C$. We use a 5-dimensional and 10-dimensional $Z$ for *Cars3D* and *Small*Norb, respectively. In Appendix A, we sweep over the dimensionalities of $C$ and $Z$ to evaluate its effect on the reported results. We train $\beta$-TCVAE with $\beta$ values ranging from 1 to 10. In addition to the main models, we include comparisons to a recent state-of-the-art method, Jacobian Supervision (Lezama, 2018). Although Jacobian supervision can be implemented with multiple student autoencoders, we only compare to the single student model version as it is most directly comparable to MS-VAE. We utilize the same architecture and $C, Z$

dimensionalities as MS-VAE. For their teacher model, we also use a pre-trained $\beta$-TCVAE model as we did for MS-VAE, but we only evaluate their method for $\beta = 1, 10$. The hyperparameters and loss functions are as specified in the paper and are available in the linked code repository.

**Measuring Disentanglement Performance**   Locatello et al. (2018) concluded that most metrics for measuring disentanglement performance correlate relatively well with each other. Therefore, we only use MIG (Chen et al., 2018) to quantitatively measure disentanglement performance. Please see Appendix G for how this is computed.

**Measuring Reconstruction Quality.**   Following (Razavi et al., 2019), we report the Frechet Inception Distance (FID) (Heusel et al., 2017) between the dataset and each models' reconstructions. We choose FID instead of reconstruction error for our main metric as L1/L2 metrics are known to not correlate well with perceived image quality (Wang et al., 2003; 2004). We still report both L1 and L2 results in Appendix J. Additionally, FID does not require a well-defined likelihood function and can, therefore, be used to evaluate MS-VAE when the conditional model is implemented using a GAN.

**Measuring the Conditioning on $C$.**   To quantitatively measure how well the improved reconstructions are conditioned on the independent latent variables $C$, we use the following procedure. Let the mean parameter of the encoder of the subgraph $C \rightarrow Y$ be $E_\phi$. Further, let $C_R = E_\phi(X)$, $C_Y = E_\phi(Y)$, $C_X = E_\phi(G_\Theta(Y, Z))$, $C_{z_\epsilon} = E_\phi(G_\Theta(Y, \epsilon)))$, and $C_{Y_\epsilon} = E_\phi(G_\Theta(\epsilon, Z)))$ where $X$ is an input image, $\epsilon \sim \mathcal{N}(0, I)$, and $G_\Theta$ is the decoder of the second stage of MS-VAE. We report the following mutual information terms: $M_1 = \text{MI}[C_R; C_Y]$, $M_2 = \text{MI}[C_R; C_X]$, $M_3 = \text{MI}[C_R; C_{z_\epsilon}]$, and $M_4 = \text{MI}[C_R; C_{Y_\epsilon}]$. If $Z$ is encoding meaningful information for the reconstruction process then we should see $M_2 > M_1$ as the reconstruction should better approximate the ground truth image X. More importantly, if $M_1 > M_3 >> M_4$ then the conditioning on $Y$ (and therefore $C$) can be assumed to be maintained. This follows because if $G_\Theta$ was only using $Z$ for the final reconstruction, $M_3$ would be close to zero and $M_4$ would be high (since Y is not being used in the final reconstruction). In addition to this quantitative analysis, we provide latent traversal plots for all the datasets to qualitatively show that the conditioning on the disentangled factors $C$ is preserved.

**Downstream Classification**   To demonstrate that $Z$ is learning meaningful residual information that is *complimentary* to $C$, we perform a set of downstream classification tasks using both $Z$ and $C$. In these tasks, we predict attributes of objects in the *Small*NORB dataset using an MLP that is trained with three different inputs: $C$, $Z$, and both $C$ and $Z$ concatenated together. Ideally, the MLP trained with both $C$ and $Z$ would achieve the highest accuracy as the inferred latent variables contain complementary information about the object attributes.

**Code and Datasets.**   We use the *disentanglement_lib* package (Locatello et al., 2018) to train all $\beta$-TCVAE models and to evaluate MIG. The datasets we use for our quantitative benchmarks are *Cars3D* (Reed et al., 2015) and *Small*NORB (LeCun et al., 2004). These datasets are a subset of those used in the large scale study of (Locatello et al., 2018) and are available (with ground truth factors) from the *disentanglement_lib* package. Additionally, we use the CelebA (Liu et al.) dataset of real face images for qualitative evaluations since the other datasets consist of only simulated images (see Appendix B). For FID evaluations, we use the standard Tensorflow (Abadi et al., 2016) implementation for all the models. We provide all the hyperparameter and architectural details for the experiments in the appendix. A reference Tensorflow implementation is also available at `https://github.com/AnonymousAuthors000/DS-VAE` in the form of a Jupyter notebook.

## 4.1   RESULTS

**Disentanglement Performance.**   As mentioned above, MS-VAE utilizes $\beta$-TCVAE to learn disentangled factors in the first stage of its training. Therefore, we report the MIG values for the $\beta$-TCVAE used with MS-VAE and for $\beta$-TCVAE-L (a $\beta$-TCVAE with the same latent dimension as MS-VAE). For the *Cars3D* dataset, we find that with both models, the MIG scores increase as $\beta$ is increased (Figure 4a(ii)). However, $\beta$-TCVAE appears to consistently have a higher MIG than $\beta$-TCVAE-L, potentially reflecting the entangling effect of an overly large latent space. For the *Small*NORB dataset, MIG decreases as the value of $\beta$ increases as shown in Figure 4b(ii). This confirms the findings of Locatello et al. (2018) on this particular dataset. Overall, these results illustrate that MS-VAE is able to extract disentangled factors from image data at the same level as state-of-the-art methods and that naively increasing the latent dimensionality of $\beta$-TCVAE may diminish disentanglement performance. For the Jacobian supervision baseline (Lezama, 2018), we use $\beta$-TCVAE at the teacher

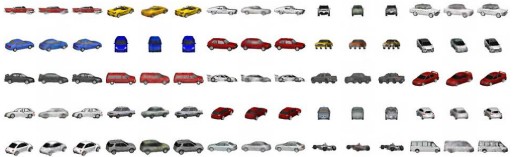
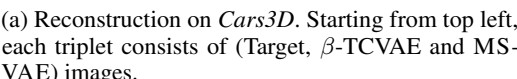

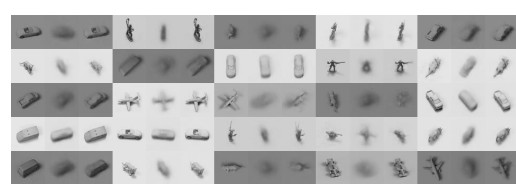

(a) Reconstruction on *Cars3D*. Starting from top left, each triplet consists of (Target, $\beta$-TCVAE and MS-VAE) images.

(b) Reconstruction on *Small*NORB. Starting from top left, each triplet consists of (Target, $\beta$-TCVAE and MS-VAE) images.

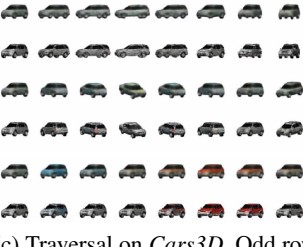

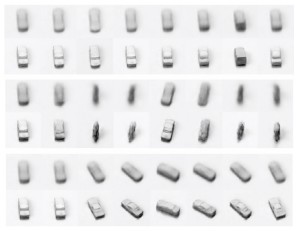

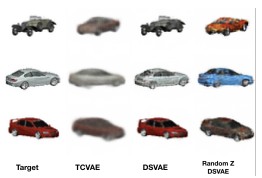

(c) Traversal on *Cars3D*. Odd row $\beta$-TCVAE, even row MS-VAE.

(d) Traversal on *Small*NORB. Odd row $\beta$-TCVAE, even row MS-VAE.

(e) Z perturbation on *Cars3D*.

Figure 3: Qualitative results on the *Cars3D* and *Small*NORB datasets for $\beta$-TCVAE and MS-VAE.

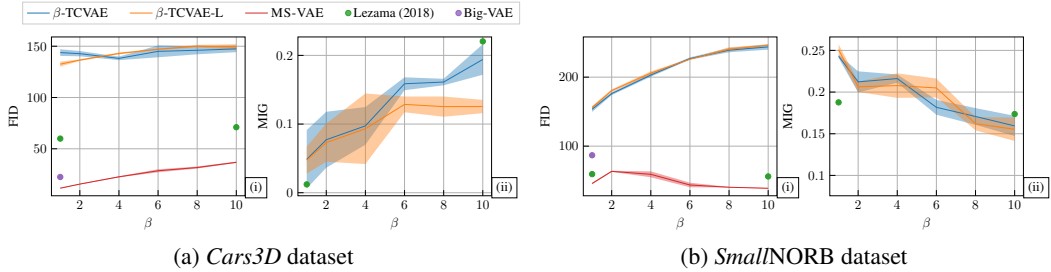

(a) *Cars3D* dataset

(b) *Small*NORB dataset

Figure 4: FID (lower is better) and MIG (higher is better) comparison of $\beta$-TCVAE, $\beta$-TCVAE-L, and MS-VAE models. On both datasets, MS-VAE is able to consistently improve the reconstruction quality of its underlying $\beta$-TCVAE model while achieving a better MIG than $\beta$-TCVAE-L . We also provide FID and MIG results for Lezama's model (Lezama, 2018) with $\beta = 1, 10$ as well as FID for a vanilla VAE model of the same capacity as MS-VAE (denoted Big-VAE).

model with two values of $\beta$ (1 and 10). After training, the MIG values for the student network are approximately the same as teacher model on both datasets (i.e. it maintains disentanglement).

**Reconstruction Quality.** As shown in Figures 4a(i) and 4b(i), MS-VAE drastically improves the reconstruction quality compared to both $\beta$-TCVAE models, achieving a much lower FID across all values of $\beta$. We also compare MS-VAE to Jacobian supervision and find that MS-VAE has lower FID on both datasets. To ensure that the improvement in reconstruction quality over $\beta$-TCVAE is not just because MS-VAE has more parameters, we trained several $\beta$-TCVAE models with the same number of parameters as MS-VAE on the *Small*NORB dataset and reported the results in Appendix H. Although we found marginal improvements in the average FID for these bigger $\beta$-TCVAEs ($145.79 \pm 2.98$), they are still far higher than MS-VAE's FID ($59.16 \pm 4.52$). Also, increasing the parameters of $\beta$-TCVAE led to a decreased MIG ($0.163 \pm 0.026$) which indicates that increasing the parameters of $\beta$-TCVAE may increase reconstruction quality at the cost of disentanglement. Finally, for reference, we provide FID scores for a standard VAE with same number of parameters as MS-VAE (we refer to this model as Big-VAE). Surprisingly, MS-VAE has a lower FID than Big-VAE at low values of $\beta$ suggesting that the residual modeling in MS-VAE leads to better image quality in general (see Figures 4a(i) and 4b(i)).

**Conditioning on $C$** Consider Figures 5a and 5b where we report the mutual information terms $M_1, M_2, M_3$, and $M_4$ for both datasets (see Section 4 for their definitions). As per our expectation,

we find that $M_2 > M_1 > M_3 >> M_4$ and that $M_3$ tracks $M_1$ without dropping significantly. This suggests that $Z$ is encoding meaningful information for the reconstruction of $X$ and that the model maintains the conditioning on $C$ in the final reconstruction. We further illustrate the conditioning on the $C$ with qualitative examples. In Figures 3(c), 3(d) and 3(e), we show latent traversal of $Y$ and $X$ pairs. Notice, MS-VAE is simply improving the quality of the generated output without diminishing the ability of the model to manipulate single factors of variation by traversing $C$.

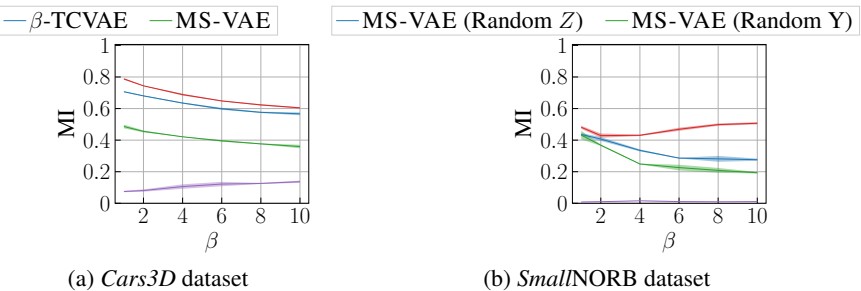

(a) *Cars3D* dataset                                    (b) *Small*NORB dataset

Figure 5: Mutual Information (MI) between inferred independent factors from the true image X (using $\beta$-TCVAE) and independent factors from various reconstructions of X. Please note that Blue = $M_1$, Red = $M_2$, Green = $M_3$, and Purple = $M_4$ (see Section 4 for their definitions).

**Downstream Classification**   The results of the downstream attribute prediction tasks are shown in Figure 6. As can be seen, the MLP that was trained with both $C$ and $Z$ attains a higher accuracy on all five attribute prediction tasks than the MLPs trained with only $C$ or $Z$. Notice that as the regularization of the latent space increases (larger $\beta$) and the accuracy of the classier trained on $C$ decreases, the accuracy of the classifier trained on $Z$ increases. This highlights that $Z$ contains complementary information to that of $C$ and sheds light on why simply increasing the size of the latent space (i.e. the $\beta$TCVAE-L model) does not improve the reconstruction significantly.

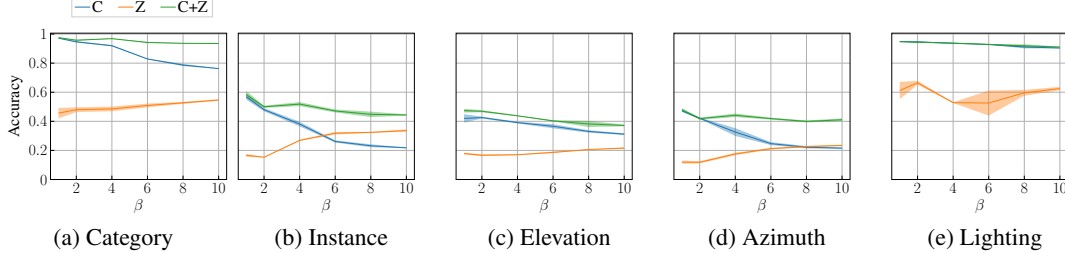

(a) Category          (b) Instance          (c) Elevation          (d) Azimuth          (e) Lighting

Figure 6: Object property prediction for *Small*NORB using inferred representations $C$, $Z$, and $C + Z$ as input to a MLP. The accuracy is highest for the MLP trained with $C + Z$.

**MS-GAN and MS-FLOW.**   We implement a GAN-based version of our framework, MS-GAN, and carry out a large battery of qualitative experiments in the supervised setting using graphics rendering programs (Appendix E). We also evaluate a simple a FLOW-based version of our framework, MS-FLOW, in Appendix D that utilizes a mixture of Gaussians model to realize the sub-graph $C \to Y$, further demonstrating the method's flexibility. As we are limited for space, we include these results in the appendices.

**Simple Pendulum.**   To demonstrate the ability of MS-VAE to learn richer and more structured hierarchies of disentangled latent factors, i.e. Deeper MS-VAE, we learn to approximate a simple pendulum simulator using MS-VAE. We refer the reader to Appendix C for the results.

## 5   CONCLUSION

In this work, we proposed a novel graphical model for DR learning that, by virtue of being D-separated, overcomes the trade-off between reconstruction and disentanglement. To implement this graphical model, we introduced a multi-stage deep generative model, MS-VAE, that learns

the independent and correlated latent variables in two separate stages of training. We showed that this model dramatically improves the reconstruction of preexisting DR learning methods while maintaining the same level of disentanglement. Furthermore, our experiments for MS-GAN and MS-FLOW (Appendices E and D, respectively) demonstrated that our framework is agnostic to both the level of supervision (e.g. unsupervised, weakly supervised, fully supervised) and to the underlying model (e.g. VAEs, GANs, normalizing flows). Finally, through the pendulum experiment (Appendix C), we showed how our framework can be extended to models with deeper and/or richer hierarchical structure. In future work, we seek to explore these deeper extensions of MS-VAE.

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

## A    CHOOZING $C$ AND $Z$

We study the impact of the dimensionality of $C$ and $Z$ on the reported results in this section. To do so, we fix a value of $\beta$ and sweep over multiple values for $C$ and $Z$.

**Choosing C:** For $C$, we sweep over the range $[1 - 10, 20, 50]$ for both the datasets. Figure 7a and 7b show the MIG and ELBO for the same. We can see that $|C| = 10$ achieves a good combination of MIG and ELBO for the *Cars3D* dataset. While there is high variance in the MIG for *Small*NORB, $|C| = 10$ performs reasonably well. This is also consistent with the large scale study in (Locatello et al., 2018), which suggests that a value of 10 for the dimensionality of $C$ is optimal for DR learning.

**Choosing Z:** Similarly for $Z$, we sweep over a set of candidate values $[5, 10, 50, 90]$ and evaluate the FID and MIG. For evaluating the FID, we consider both reconstructions (by sampling $Z$ from the posterior) and 'Random Z' (by sampling a random $Z$ from the prior). We also look at the MI G. We find smaller values, 5 and 10 for *Cars3D* and *Small*NORB, respectively, achieve a good combination of all the three quantities of interest. The results are shown in **??**.

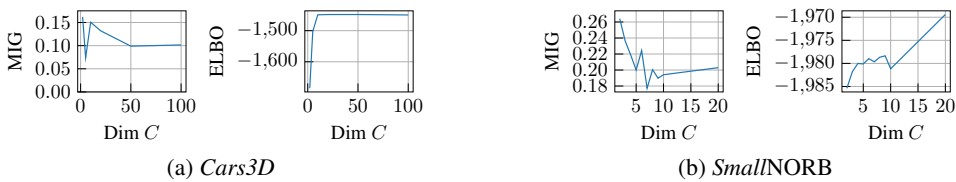

(a) *Cars3D*                    (b) *Small*NORB

Figure 7: MIG and ELBO for different dimensionality of $C$ for *Cars3D* and *Small*NORB at $\beta = 4$

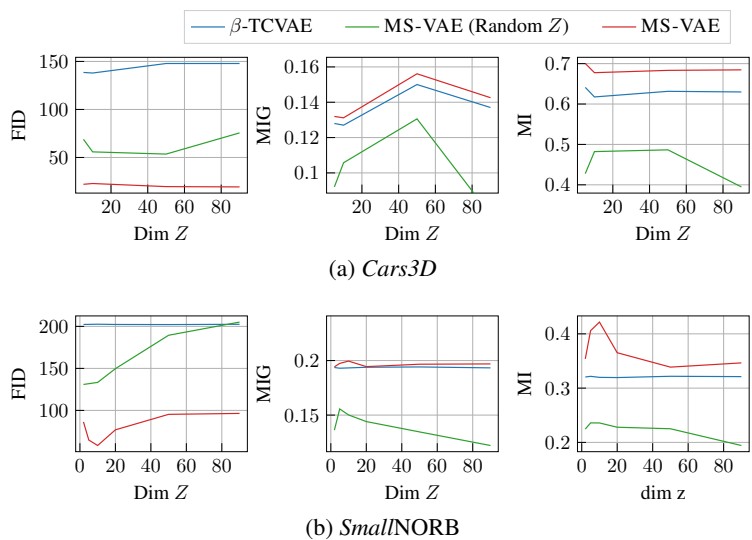

Figure 8: MIG,FID and MI plots for MS-VAE as a function of the dimensionality of $Z$ for *Cars3D* and *Small*NORB ($\beta = 4$).

Qualitatively, Figure 9 and Figure 10 show the effect of varying the dimensionality of $C$ and $Z$. For a small $C$ (Figure 9), the underlying disentanglement method does a poor job at disentangling the meaningful latent variables, which further results in blurry images. In this case, $Z$ has substantial control over the image—it adds various missing details to $Y$ such as color. In contrast, Figure 10 shows traversals for a high value of $C$. Now, each individual dimension of C controls fewer independent factors, the $Y$ images are relatively sharper, and adding $Z$ improves them further. Similar conclusions hold for *Small*NORB as shown in Fig. 12 and 11.

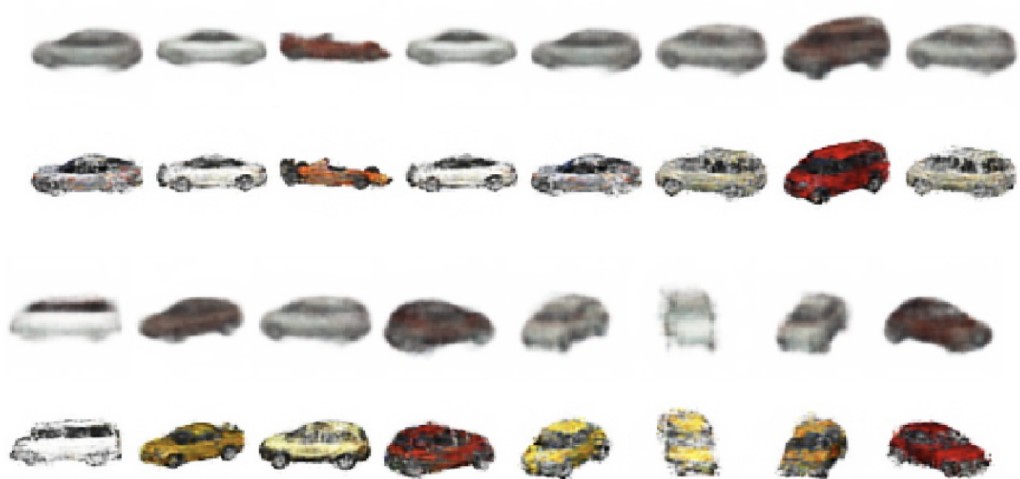

Figure 9: Traversals for cars for small $C$ (c = 2, z = 8 and $\beta$ = 4). Odd rows $\beta$-TCVAE and even rows MS-VAE. We can see the $Y$ images are blurry and various factors are entangled.

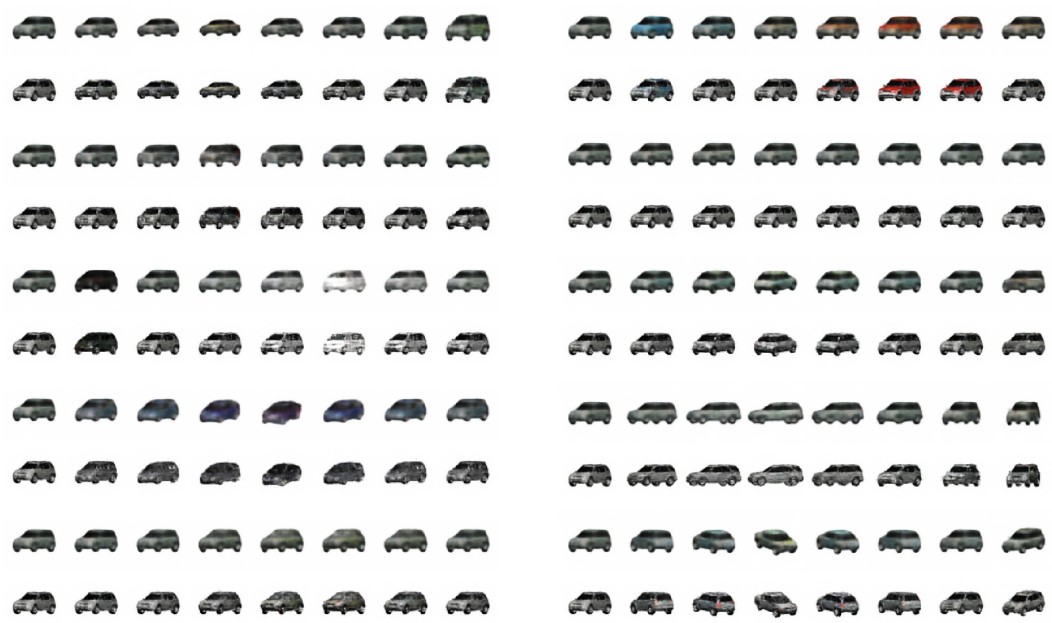

Figure 10: Traversals for cars large $C$ (c = 10, z = 5 and $\beta$ = 4). Odd rows $\beta$-TCVAE and even rows MS-VAE. Larger $C$ achieves greater disentanglement and $Z$ further refines the images.

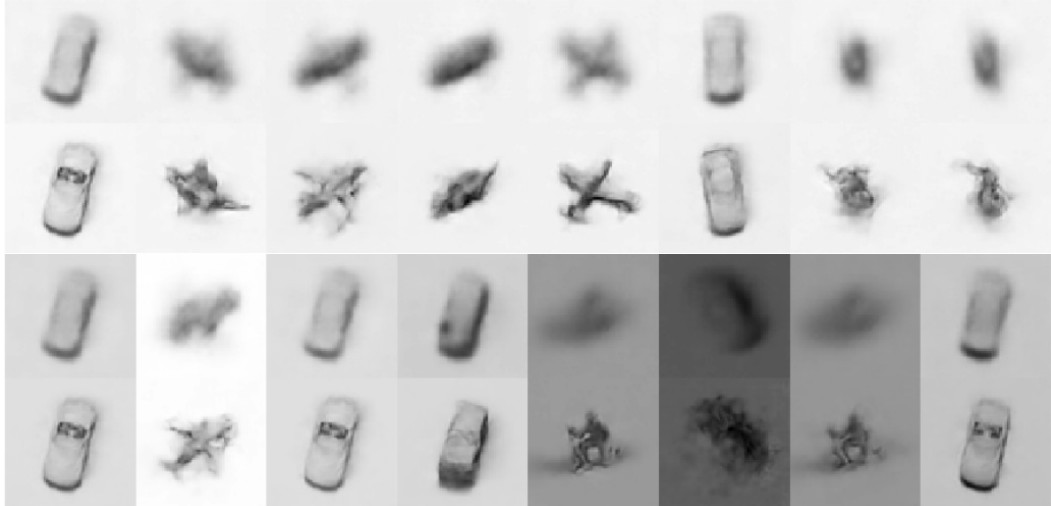

Figure 11: Traversals for SmallNORB c = 2, z = 8 and $\beta$ = 4. Odd rows $\beta$-TCVAE and even rows MS-VAE.

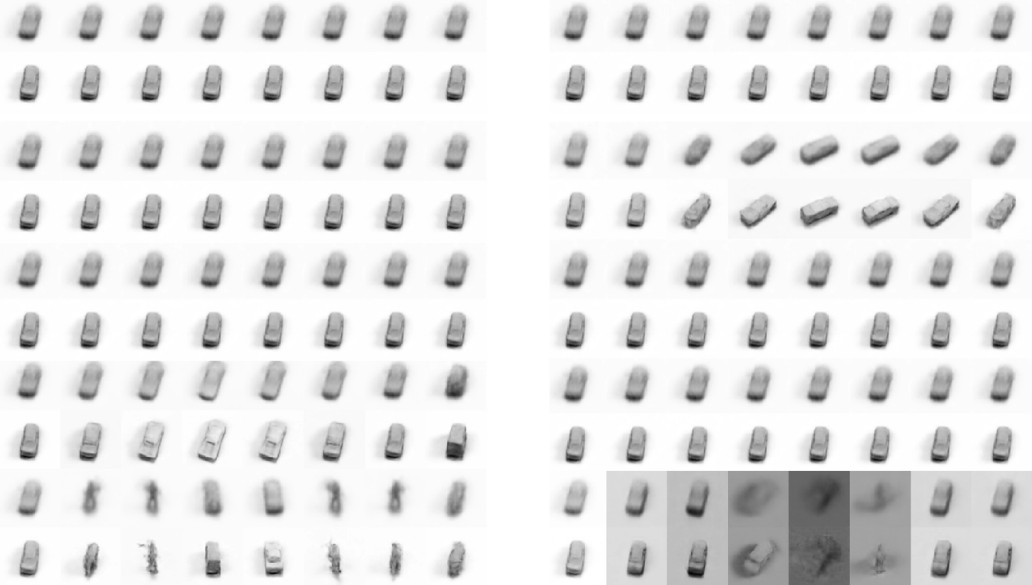

Figure 12: Traversals for SmallNORB c = 10, z = 10 and $\beta$ = 6. Odd rows $\beta$-TCVAE and even rows MS-VAE.

## B    CELEBA RESULTS

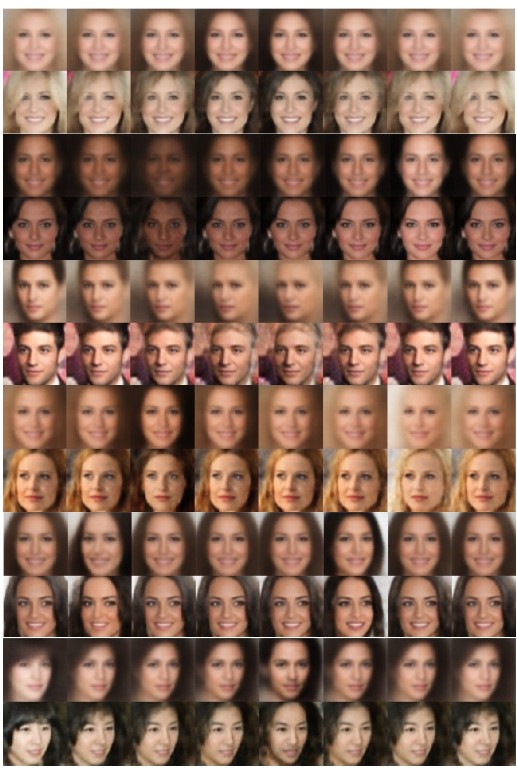

Figure 13: CelebA Latent traversal on $C$. Odd rows are $\beta$-TCVAE and even rows are MS-VAE.

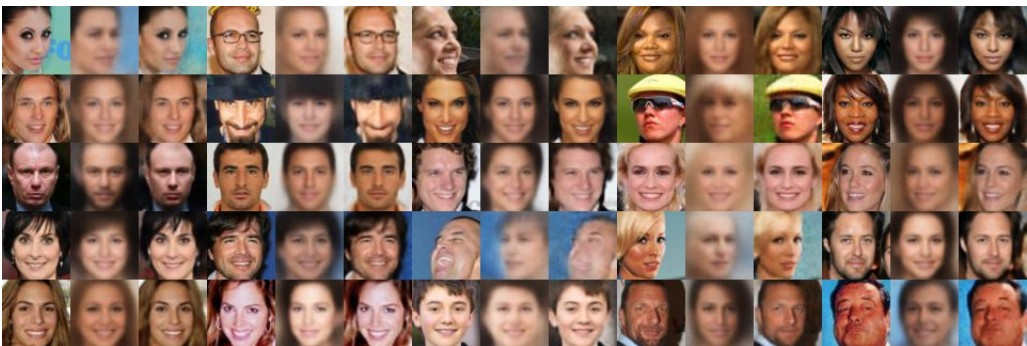

Figure 14: CelebA Reconstruction plots.

## C  MS-VAE FOR SIMPLE PENDULUM

A simple, 2D pendulum is comprised of a bob with mass $m$ attached to a string of length $L$ where the mass of the string is negligible in comparison to $m$. The primary forces acting upon the bob are the gravitational force, the tension force of the string, and the damping force from air resistance. Let $\theta$ be the angular displacement of the weight (the corresponding angle with respect to the y-direction). The pendulum's motion can be completely described as $\frac{d^2\theta}{dt^2} = -\frac{b}{m}\frac{d\theta}{dt} - \frac{g}{L}\sin\theta$, where $b$ is the damping coefficient and $g$ is the gravitational acceleration constant (the initial angular displacement/velocity are constant).

In this toy example, we assume that, given the length $L$, the damping coefficient $B$, and the mass $M$, the angular displacement of a pendulum is generated by a noiseless simulator $S : (L, M, B) \rightarrow \theta$ where $\theta := \{\theta_0, \theta_1, ..., \theta_T\}$, $\theta_i \in [-\pi, \pi]$, and $T = 100$. Using samples from $S$ where only $L$ and $B$ are known a priori, we aim to learn a hierarchical, generative model for the pendulum where *all latent variables are completely disentangled*. With MS-VAE, we can achieve this by learning the graphical model in Figure 17. In this graphical model, we introduce two intermediate observed nodes $Y_0$ and $Y_1$ that block the path between $L$, $B$, and $Z$. We define $Y_0$ as the undamped angular displacement and $Y_1$ as the damped angular displacement with fixed mass ($M = 1$). This simple example demonstrates how the MS-VAE framework can be extended to models with more structured hierarchies while consistently satisfying the conditional independence assumptions implied by the simulator.

We train our model in three separate steps. First, we use paired data $\{L^i, y^i_0\}$ to learn the subgraph $L \rightarrow Y_0$ as a conditional Gaussian distribution (how the pendulum length affects the angular displacement). Second, we use the paired data $\{b^i, y^i_0, y^i_1\}$ to learn $(B, Y_0) \rightarrow Y_1$ as a conditional Gaussian distribution (how the damping coefficient affects the angular displacement for a fixed mass). Finally, we use the paired data $\{y^i_1, x^i\}$ to learn the subgraph $(Z, y^i_1) \rightarrow \theta$, where $Z$ models the residual between $Y_1$ and the true observations $\theta$ (this subgraph is realized as a VAE). Through construction and training, the latent variable $L$ should control the length of the pendulum, the latent variable $B$ should control the damping coefficient, and the latent variable $\boldsymbol{Z}$ should be correlated with the mass $M$. [1] Note that while we have labeled data for the latent variables $L$ and $B$, we never have access to the latent variable $M$. Despite this, the latent variable $Z$ can still learn to model the effect of $M$ by modeling the residual.

The results are shown in Figures 15 and 16. As can be seen, MS-VAE is able to learn completely disentangled latent variables using a combination of supervision and hierarchical modeling. The final learned latent variable Z is shown to be heavily correlated with the mass of the pendulum, highlighting how MS-VAE's learned latent variables can be directly related with informative physical parameters. This simple toy example highlights two powerful features of MS-VAE. First, the hierarchy of latent variables can be made arbitrarily deep. Second, our method can be used to discover interpretable latent factors when only some of control variates $C$ are known a priori (e.g. the length $L$ and damping coefficient $B$).

---

[1] For simplicity, we don't model the data as time series explicitly but rather as a fixed length, 1D vector.

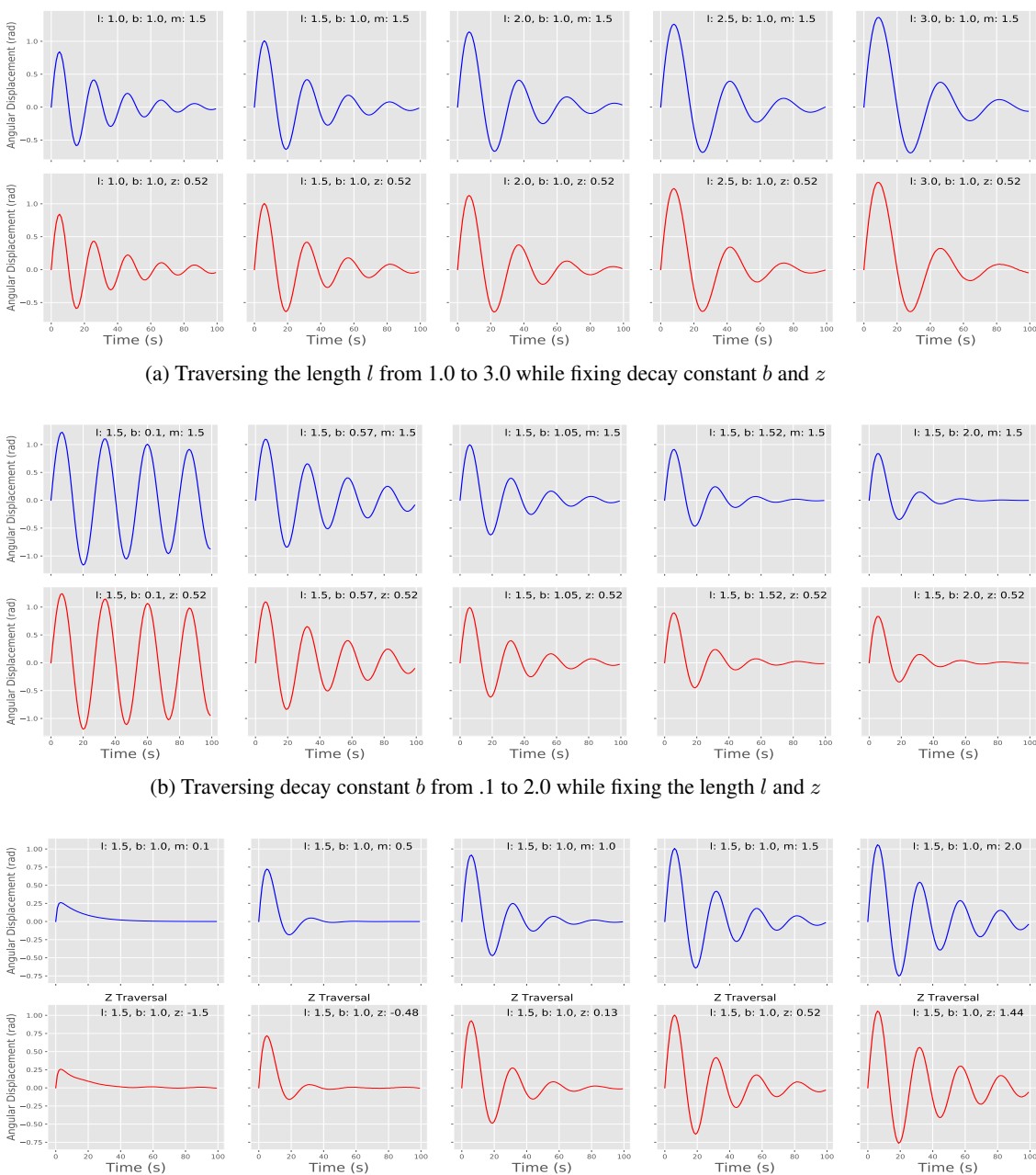

(a) Traversing the length $l$ from 1.0 to 3.0 while fixing decay constant $b$ and $z$

(b) Traversing decay constant $b$ from .1 to 2.0 while fixing the length $l$ and $z$

(c) Traversing the mass $m$ from .1 to 2.0 (top) and traversing the corresponding $z$ values (-1.5 to 1.44) while fixing the length $l$ and decay constant $b$.

Figure 15: Latent traversals for the pendulum model. Odd rows (blue) are actual data from the simulator and even rows (red) are from MS-VAE. In this example, MS-VAE learns each of the three latent variables in a disentangled manner without any supervision for the mass control variate. We choose the $z$ values that are plotted by examining the posterior over $z$ and choosing $z$ values that correspond to the given masses ($z$ is 1-Dimensional so this is straightforward).

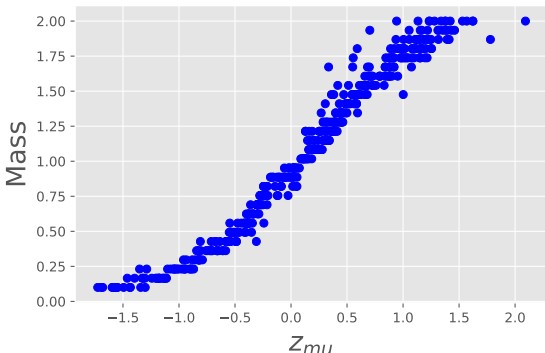

Figure 16: Correlation between $z$ and the mass when modeling a simple pendulum. In this plot, we show 500 1-Dimensional posterior means for different $\theta$ observations to illustrate how mass and the latent variable Z are heavily correlated after training.

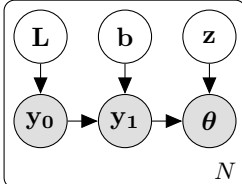

Figure 17: MS-VAE graphical model of a simple pendulum. $\theta$ is the angular displacement; $L$ is the length; $B$ is the damping coefficient. After learning and by construction, the latent variable Z is correlated with the pendulum's mass.

## D    FLOW-BASED IMPLEMENTATION (MS-FLOW)

Normalizing flows are powerful and flexible parametric distributions which are also tractable (Dinh et al., 2014; Huang et al., 2018; Durkan et al., 2019); for a review see Kobyzev et al. (2019). The conditional model in MS-VAE $p_\Theta(x|z, y)$ can also be implemented as a conditional normalizing flows (that conditions on $Y$) instead of a VAE.

As an illustration, in this section we present the MS-FLOW model on the MNIST digit dataset. Here the subgraph $C \to Y$ is realized as a Gaussian mixture model and the subgraph $(Y, Z) \to X$ is implemented as a conditional coupling flow (Dinh et al., 2016). We fit a mixture of 10 Gaussians whose means are shown in Figure 18. Figure 19 shows the real data, samples from the Gaussian

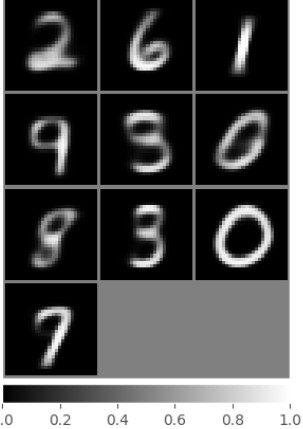

Figure 18: The means of the Gaussian mixture part of MS-FLOW on MNIST.

mixture and the conditional flow. As expected, the log density of the overall model improves from -1585.086 to -1356.089 once the FLOW is applied to the output of the mixture model. All while maintaining the ability of controllable sampling.

### D.1    TRAINING USING MEANS OR SAMPLES

As mentioned in previous sections, when training the conditional model, we can use either the means from the likelihood model of the $C \to Y$ sub-graph, or the samples from it. We fit the MS-FLOW model with both strategies and show the change of log density during training in Figure 20 Using the means makes the training converge much faster than using the samples. It is not surprised at all, because using samples leads to noisy gradients during training. Thus, for faster convergence, we use means rather than samples in all of our experiments, even though it make the estimates slightly biased.

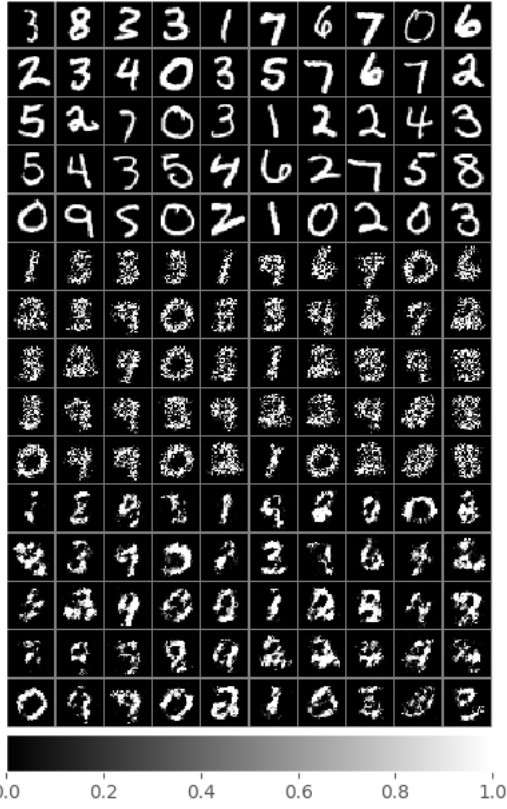

Figure 19: MS-FLOW on MNIST. First 5 rows are real data, next five rows are samples from the Gaussian mixture and last five rows are from the conditional flow.

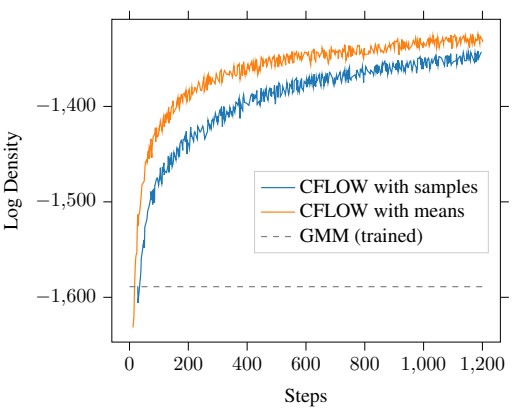

Figure 20: Conditional log density during training for different types of $Y$ (samples v.s. means).

## E    GAN-BASED IMPLEMENTATION (MS-GAN)

Equation 2 also allows for likelihood-free inference with a little manipulation. Simply by adding and subtracting the log of the true conditional density $\log p(X|Y)$ in equation 2, we get the lower bound,

$$\log p_{\{\phi,\theta,\theta_Z,\Theta\}}(x,y) \geq -\mathcal{KL}[q_\phi(c|y)\|p(c)]+$$

$$\int q_\phi(c|y) \log p_\theta(y|c)dc - \mathcal{KL}[q_{\theta_Z}(z|x,y)\|p(z)]+$$

$$\underbrace{\int q_{\theta_Z}(z|x,y) \log \frac{p_\Theta(x|z,y)}{p(x|y)}dz}_{\text{GAN-based }(Y,Z)\to X} + \log p(x|y) \tag{3}$$

Here, the log density ratio $\log \frac{p_\Theta(x|z,y)}{p(x|y)}$ can be estimated using adversarial learning with a binary discriminator function (Sugiyama et al., 2012; Srivastava et al., 2017). In practice, since we are learning conditional distributions, the decoder is implemented as a conditional GAN Mirza & Osindero (2014) This implementation is referred to as MS-GAN.

### E.1    PRELIMINARIES: GENERATIVE ADVERSARIAL NETWORKS

Generative adversarial networks (GAN) (Goodfellow et al., 2014) represent the current state of the art in likelihood-free generative modeling. In GANs, a generator network $G_\theta$ is trained to produce samples that can fool a discriminator network $D_\omega$ that is in turn trained to distinguish samples from the true data distribution $p(x)$ and the generated samples $G_\theta(z)|z \sim p_z(z)$. Here, $p_z$ is usually a low dimensional easy-to-sample distribution like standard Gaussian. A variety of tricks and techniques need to be employed to solve this min-max optimization problem. For our models, we employ architectural constraints proposed by DC-GAN (Radford et al., 2015) that have been widely successful in ensuring training stability and improving generated image quality.

Conditional GANs (CGAN) (Mirza & Osindero, 2014) adapt the GAN framework for generating class conditional samples by jointly modeling the observations with their class labels. In CGAN, the generator network $G_\theta$ is fed class labels $c$ to produce fake conditional samples and the discriminator $D_\omega$ is trained to discriminate between the samples from the joint distribution of true conditional and true labels $p(x|c)p(c)$ and the fake conditional and true labels $p_\theta(x|c)p(c)$.

While not the main focus of this paper, we present a novel information theoretic perspective on CGANs. Specifically, we show that CGAN is trained to maximize a lower-bound to the mutual information between the observation and its label while simultaneously minimizing an upper-bound to it. We state this formally:

**Lemma 1** (Information-theoretic interpretation of CGAN). *Given* $(x,c) \sim p(x,c)$*, CGAN learns the distribution* $p_\theta(x) = G_\theta(x)$ *by training a discriminator* $D_\omega$ *to approximate the log-ratio of the true and generated data densities i.e.* $D_\omega \approx \log p(x,c)/p_\theta(x,c)$ *in turn minimizing the following*

$$\min_\theta \mathbb{E}_{p_\theta(x|c)p(c)}\big[-D_\omega\big]$$

$$\approx \min_\theta \big(\mathcal{I}_{g,\theta}(x,c) + \mathbb{E}_{q(c|x,\theta)}KL(p_\theta(x)\|p(x))\big)+$$

$$\max_\theta \big(\mathcal{I}_{g,\theta}(x,c) - \mathbb{E}_{p_\theta(x)}KL[q(c|x,\theta)\|p(c|x)]\big)$$

$$= \min_\theta \mathcal{I}_{g,\theta}^{UB}(x,c) - \mathcal{I}_{g,\theta}^{LB}(x,c).$$

*where* $\mathcal{I}_{g,\theta}(x,c)$ *is the generative mutual information and* $q(c|x,\theta)$ *is the posterior under the learned model.*

The detailed derivation is provided in the section below. Notice that at the limit, the model learns exactly the marginal distribution of $x$ and the posterior $q(c|x)$ and the KL terms vanish.

### E.2    CGAN MUTUAL INFORMATION DERIVATION

Following Sugiyama et al. (2012); Gutmann & Hyvärinen (2010); Mohamed & Lakshminarayanan (2016); Srivastava et al. (2017) we know that at optima the **logits** ($D_\omega$) of a trained discriminator

approximate the log-ratio of the true data and generated data densities, i.e. $D_\omega \approx \log \frac{p_x(x|c)p(c)}{p_\theta(x|c)p(c)}$. Following Nowozin et al. (2016); Srivastava et al. (2017; 2018; 2019) the generator $G_\theta$ can therefore be trained to minimize the following f-divergence between the two sets of densities,

$$\min_\theta \mathbb{E}[-D_\omega]$$
$$\approx \min_\theta \int p_\theta(x|c)p(c)\left[-\log\frac{p(x|c)p(c)}{p_\theta(x|c)p(c)}\right]d(x,c). \tag{4}$$

The RHS of equation 4 can be re-arranged into terms containing the upper and the lower bounds to the generative mutual information between $X$ and $C$, i.e.,

$$\min_\theta \mathbb{E}\big[-D_\omega\big]$$
$$\approx \min_\theta \mathbb{E}\left[\log\frac{p_\theta(x|c)}{p(x)}\right] + \max_\theta \mathbb{E}\left[\log\frac{p(c|x)}{p(c)}\right]$$
$$= \min_\theta \mathbb{E}\left[\log\frac{p_\theta(x|c)}{p_\theta(x)} + \log\frac{p_\theta(x)}{p(x)}\right] +$$
$$\max_\theta \mathbb{E}\left[\log\frac{q(c|x,\theta)}{p(c)} + \log\frac{p(c|x)}{q(c|x,\theta)}\right]$$
$$= \min_\theta \mathbb{E}\left[\log\frac{p_\theta(x|c)}{p_\theta(x)} + \log\frac{p_\theta(x)}{p(x)}\right] +$$
$$\max_\theta \mathbb{E}\left[\log\frac{q(c|x,\theta)}{p(c)} - \log\frac{q(c|x,\theta)}{p(c|x)}\right]$$
$$= \min_\theta \mathcal{I}_{g,\theta}(x,c) + \mathbb{E}_{q(c|x,\theta)}\mathrm{KL}(p_\theta(x)\|p(x))+$$
$$\max_\theta \mathcal{I}_{g,\theta}(x,c) - \mathbb{E}_{p_\theta(x)}\mathrm{KL}[q(c|x,\theta)\|p(c|x)]$$
$$= \min_\theta \mathcal{I}_{g,\theta}^{UB}(x,c) - \mathcal{I}_{g,\theta}^{LB}(x,c). \tag{5}$$

### E.3 LEARNING $(Y, Z) \to X$ USING CGAN

Vanilla GANs can only model the marginal data distribution i.e. they learn $p_\theta$ to match $p_x$ and in doing so they use the input to the generator ($G_\theta$) only as a source of stochasticity. Therefore we start with a conditional GAN model instead, to preserve the correspondence between $Y$ and $X$. As shown in section E.1, this framework trains $G_\theta$ such that the observation $X$ is maximally explained by the conditioning variable $Y$. One major deviation from the original model is that the conditioning variable in our case is the same type and dimensionality as the observation. That is, it is an image, albeit a blurry one. This setup has previously been used by Isola et al. (2017) in the context of image-to-image translation.

Incorporating $Z$ requires careful implementation due to two challenges. First, trivially adding $Z$ to the input along with $Y$ invokes d-separation and as a result $Y$ and $Z$ can get entangled. Intuitively, $Z$ is adding high level details to the intermediate representation $Y$. We leverage this insight as an inductive bias, by incorporating $Z$ at higher layers of the network rather than just feeding it as an input to the bottom layer. A straightforward implementation of this idea does not work tough. The reason is that BatchNorm uses batch-level statistics to normalize the incoming activations of the previous layer to speed up learning. In practice, mini-batch statistics is used to approximate batch statistics. This adds internal stochasticity to the generator causing it to ignore any externally added noise, such as $Z$. An elegant solution to resolve this second challenge comes in the form of adaptive instance normalization (see Section 3.2). It not only removes any dependency on the batch-statistics but also allows for the incorporation of $Z$ in the normalization process itself. For this reason, it has previously been used in style transfer tasks (Huang & Belongie, 2017). We replace all instances of BatchNorm in the generator with Adaptive InstanceNorm. We then introduce $Z$ to the generative process using equation **??**. $\gamma(z)$ and $\beta(z)$ are parameterized as a simple feed-forward network and are applied to each layer of AdaIN in the generator.

## F    EXPERIMENTS FOR MS-GAN

In this section, we provide a comprehensive set of qualitative results to demonstrate how MS-GAN is clearly able to not only disentangle $C$ from $Z$ in both supervised and unsupervised settings but also ensure that independent components of $C$ stay disentangled after training. Additionally, we show how in unsupervised settings MS-GAN can be used to discover disentangled latent factors when $C$ is not explicitly provided.

We evaluate MS-GAN on a variety of image generation tasks which naturally involve observed attributes $C$ and unobserved attributes $Z$. To that end, we generate three 3D image datasets of faces, chairs, and cars with explicit control variables. Chairs and cars datasets are derived from ShapeNet (Chang et al., 2015). We sample 100k images from the full yaw variation and a pitch variation of 90 degrees. We used the straight chair subcategory with 1968 different chairs and the sedan subcategory with 559 different cars. We used Blender to render the ShapeNet meshes scripted with the Stanford ShapeNet renderer. For faces, we generated 100k images from the Basel Face Model 2017 (Gerig et al., 2018). We sample shape and color (first 50 coefficients), expressions (first 5 coefficients), pose (yaw -90 to 90 degrees uniformly, pitch and roll according to a Gaussian with variance of 5 degrees) and the illumination from the Basel Illumination Prior (Egger et al., 2018). For the generation of the faces dataset, we use the software provided by Kortylewski et al. (2019). For the stated datasets we have complete access to $C$, however, we also include unsupervised results on celebA (Liu et al., 2015) with unconstrained, real images. All our datasets are built from publicly available data and tools.

We use the DCGAN architecture  (Radford et al., 2015) for all neural networks involved in all the experiments in this work and provide a reference implementation with exact architecture and hyperparameter settings at *https://github.com/AnonymousAuthors000/DS-VAE*.

### F.1    SUPERVISED SETTING

In the supervised setting we compare MS-GAN to CGAN qualitatively. To evaluate the level of disentanglement between $C$ and $Z$, we vary each individual dimension of $C$ over its range while holding $Z$ constant. We plot the generated images for both models on car and chair datasets in Figure 21. Notice that MS-GAN allows us to vary the control variates without changing the identity of the object, whereas CGAN does not. In addition, we find that for CGAN, the noise $Z$ provides little to no control over the identity of the chairs. This is potentially due to the internal stochasticity introduced by the BatchNorm. The last rows for the MS-GAN figures provide the visualization of $Y$. It can be seen how $Y$ is clearly preserving $C$ (pose information) but averaging the identity related details.

We also qualitatively evaluate MS-GAN on the more challenging faces dataset that includes 10 control variates. As shown in Figure 25 in the appendix, MS-GAN is not only able to model the common pose factors such as rotation and azimuth but also accurately captures the principal shape component of Basel face model that approximates the width of the forehead, the width of jaw etc. Compared to CGAN, MS-GAN does a qualitatively better job at keeping the identity constant.

### F.2    UNSUPERVISED SETTING

We now test the performance of MS-GAN in the unsupervised setting, where disentangled components of $C$ needs to be discovered, using $\beta$-VAE, as part of learning the mapping $C \rightarrow Y$. For our purpose, we use a simple version of the original $\beta$-VAE method with a very narrow bottleneck (6D for faces and 2D for cars and chairs) to extract $C$.

The latent traversals for the faces dataset are presented in Figure 22. Unsupervised discovery is able to recover rotation as well as translation variation present in the dataset. For comparison, we evaluate InfoGAN (Chen et al., 2016) and present the results in Figure 23 where it is evident that MS-GAN clearly outperforms InfoGAN on both disentanglement and generative quality. More traversal results are provided in the appendix. We further test our method on the CelebA dataset (Liu et al., 2015), where pose information is not available. This traversal plot is shown in Figure 24. Traversal plots for cars and chairs dataset are provided in the Figures 28 and 29.

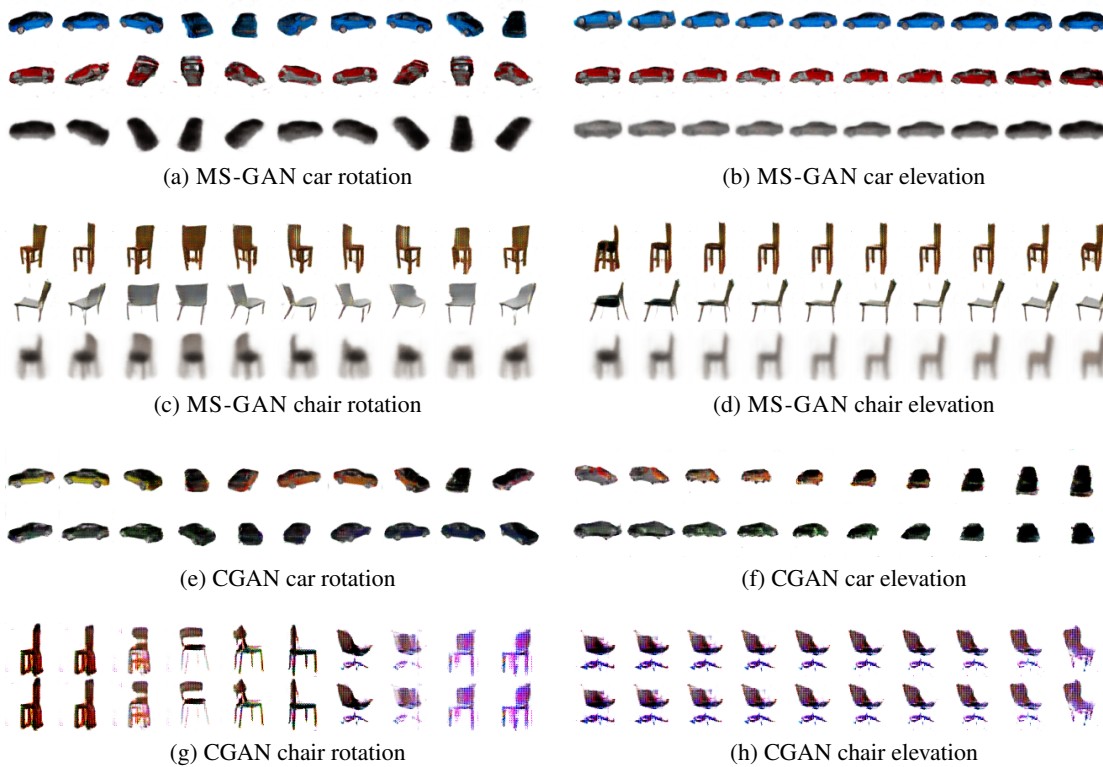

(a) MS-GAN car rotation

(b) MS-GAN car elevation

(c) MS-GAN chair rotation

(d) MS-GAN chair elevation

(e) CGAN car rotation

(f) CGAN car elevation

(g) CGAN chair rotation

(h) CGAN chair elevation

Figure 21: Latent traversal on cars and chairs. Third rows in MS-GAN results show $Y$.

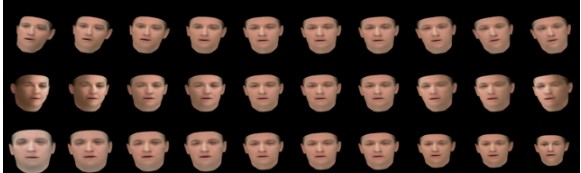

Figure 22: Latent traversal on faces (unsupervised MS-GAN). The three latent variables capture the rotation, azimuth, and distance respectively.

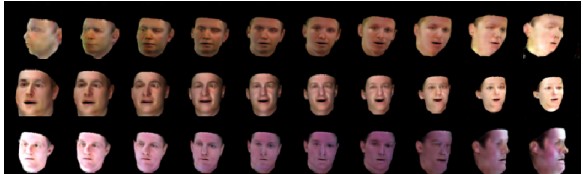

Figure 23: Latent traversal of InfoGAN on faces. The latent variables are able to capture some pose changes but the pose changes are highly entangled with other pose factors as well as the face shape.

### F.3 ADDITIONAL EXPERIMENT RESULTS

Comparison of MS-GAN and CGAN on face dataset is shown in Figure 25. CGAN not only produces blurry faces but also shows more undesired identity changes. In order to show the shape variation clearly, we provide a zoomed-in view in Figure 26.

We provide additional results for supervised and unsupervised results on the chair dataset from Aubry et al. (2014) in Figure 27 and Figure 28 respectively. The observation is the same with the previous one. MS-GAN varies the control variables without changing the shape of chairs. In the first row in

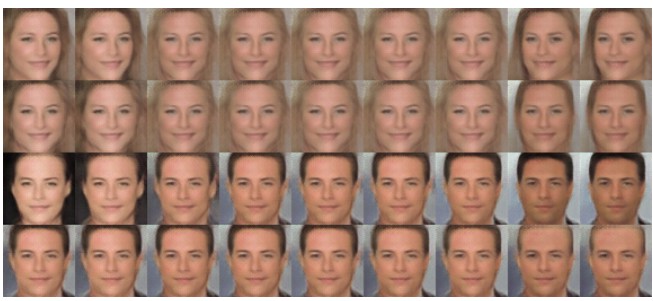

Figure 24: Latent traversal on CelebA (unsupervised MS-GAN). The latent variables consistently capture the azimuth, hair-style, gender and hair color respectively while maintaining good image quality.

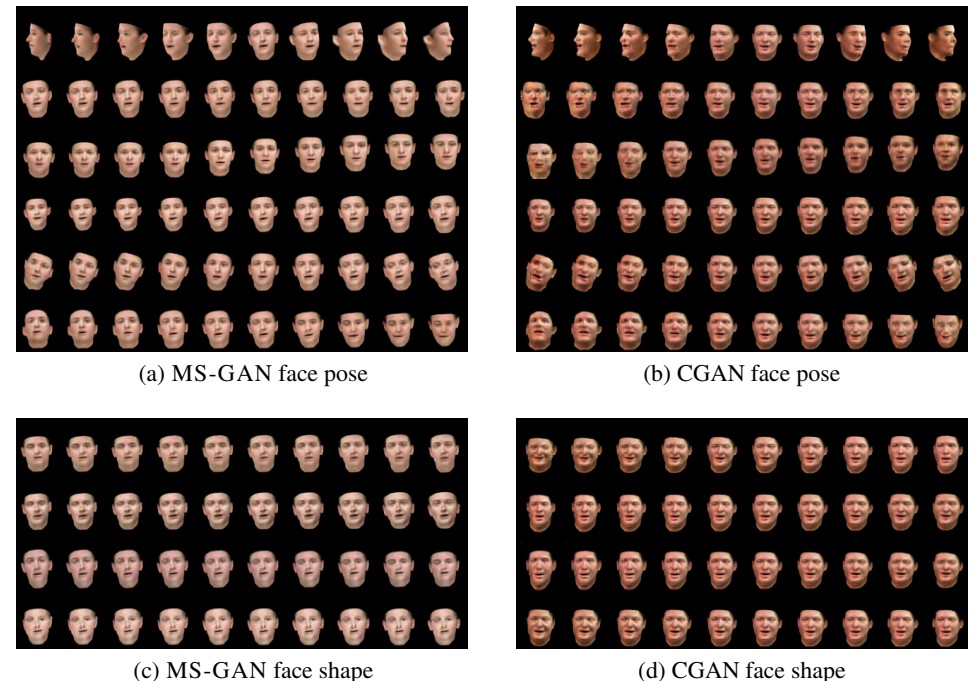

(a) MS-GAN face pose

(b) CGAN face pose

(c) MS-GAN face shape

(d) CGAN face shape

Figure 25: Latent traversal of MS-GAN and CGAN on faces. The pose variations are azimuth, horizontal translation, vertical translation, distance, rotation, and elevation from top to bottom. The shape variations show the difference in face height, forehead, jaw, and ear from top to bottom.

Figure 27, the leg of the chairs are visually indistinguishable showing an excellent disentanglement between $C$ and $Z$. For the results in unsupervised setting showing in Figure 28, MS-GAN is able to disentangle the rotation of chairs without any label.

Additional results of latent traversal of MS-GAN in the unsupervised setting is provided in Figure 29. The model is able capture the rotation but the translation is not very smooth.

Figure 30 provides the InfoGAN result on the face dataset. Compared with unsupervised MS-GAN result in Figure 31, clearly InfoGAN discovers some control variables but the effect is highly entangled.

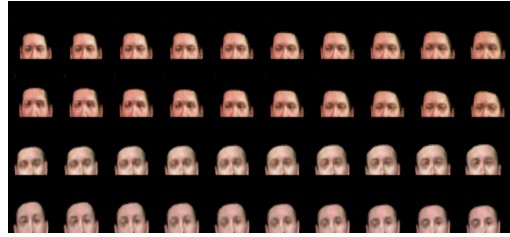

Figure 26: Zoomed-in comparison on face shape. Row 1: CGAN forehead variation; Row 2: CGAN jaw variation; Row 3: MS-GAN forehead variation; Row 4: MS-GAN jaw variation. Row 1 and Row 3 should have a bigger forehead from left to right while Row 2 and Row 4 should have a consistent forehead. CGAN and MS-GAN show good forehead variation in Row 1 and Row 3, respectively, but MS-GAN is better at keeping the forehead the same while another factor is changing (Row 4 vs. Row 1).

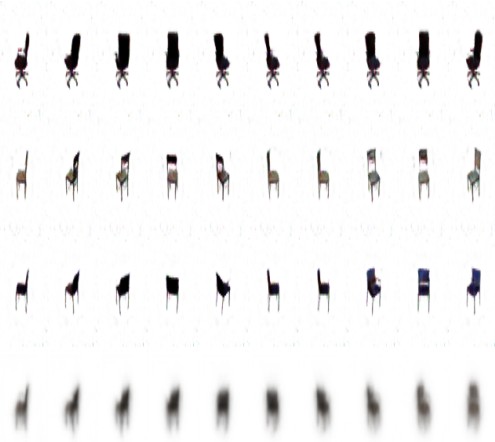

Figure 27: Latent traversal on chairs of MS-GAN. The first three rows show the effect of the variable in $C$ that controls rotation. The last row is the corresponding $Y$.

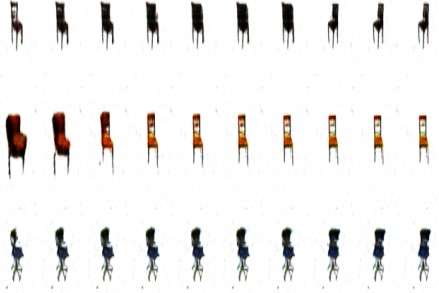

Figure 28: Latent traversal on chairs of unsupervised MS-GAN.

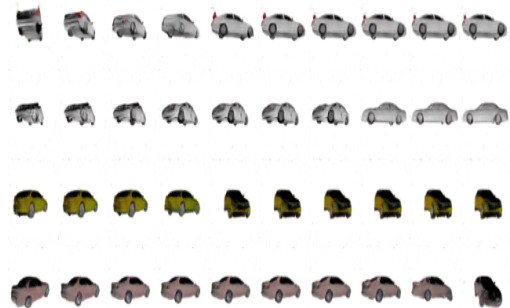

Figure 29: Latent traversal of unsupervised MS-GAN on cars.

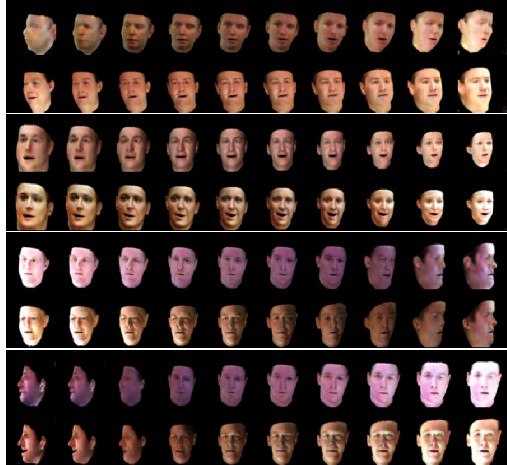

Figure 30: Latent traversal of InfoGAN on faces dataset.

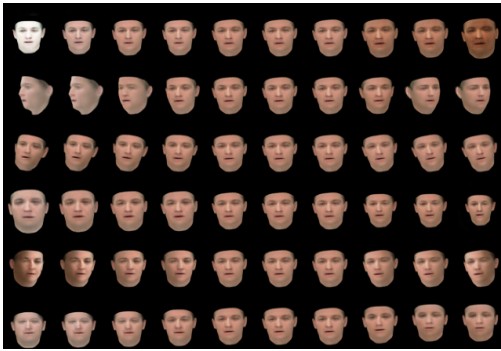

Figure 31: Latent traversal of unsupervised MS-GAN on face dataset.

## G    METRIC FORMULATION

The normalized mutual information (MI) reported in Figure 35a(iii) and Figure 35b(iii) is calculated as follows.

$$MI[C_R, C] = \frac{1}{n} \sum_{i=1}^{n} \frac{I(c^i; c_R^i)}{H(c_R^i)} \tag{6}$$

where $n$ is the dimension of $C$ and $C_R$ and $I(c^i; c_R^i)$ is the mutual information between $c^i$ and $c_R^i$.

For every ground truth factor, we compute its mutual information with each one of the learned dimensions of $C$, normalized by its entropy. For each factor, we take the difference, in mutual information, between the top two learned variables with which it has the highest mutual information. MIG is defined to be the average of that value over all ground truth factors.

# H    COMPARISON OF MS-VAE AND BIG-$\beta$-TCVAE

Table 1 shows the comparison of MS-VAE with Big-$\beta$-TCVAE which is a $\beta$-TCVAE that has the same capacity in terms of the number of parameters and the latent dimensionality. Our experiments reveal that MS-VAE outperforms the Big-$\beta$-TCVAE in terms of both disentanglement representation learning quality and reconstruction quality.

Table 1: Comparion of MS-VAE with Big $\beta$-TCVAE.

| Dataset | Model | FID | MIG |
|---------|-------|-----|-----|
| Cars | Big-$\beta$-TCVAE | 35.24±1.28 | 0.09±0.04 |
| | MS-VAE | **22.72±3.66** | **0.10±0.03** |
| *Small*NORB | Big-$\beta$-TCVAE | 152.19±2.68 | 0.16±0.02 |
| | MS-VAE | **59.16±4.52** | **0.22±0.01** |

# I    SAMPLING MULTIPLE C AND Y DURING TRAINING

In our quantitative experiments for MS-VAE, the paired training set $\mathcal{D}_{\text{paired}}$ was constructed as follows: $x^i \sim p(x)$, $c \sim q_\phi(c|x)$, $y^i = \mathbb{E}_y[p_\theta(y|c)]$. In this construction, only one $c$ and, consequentially, only one $y$ is sampled for training. For the qualitative experiments with celebA, we sampled 10 different $c$'s from the $\beta$-TCVAE for each $x_i$ such that the final dataset had 10 different $y_i$'s for each $x_i$.

The reason we sampled more $y_i$'s for each $x_i$ is that there was high variability of the sampled $y_i$'s given the same $x_i$. To visualize this effect, we plot five Y samples for five X samples from a trained $\beta$-TCVAE ($\beta = 15, C = 15$) in figure 32. As can be seen from in the figure, there is high variability in the sampled $y_i$'s for each $x_i$, highlighting the the high variance of $\beta$-TCVAE's posterior distribution over C. Training with multiple $y_i$'s for each $x_i$ allows the MS-VAE to naturally adapt to this variability as shown in 33. We plan to explore this effect in future work.

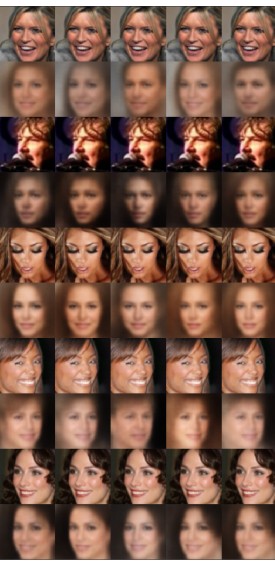

Figure 32: Even rows are samples from CelebA (the same sample is plotted five times) and odd rows are corresponding samples from the $\beta$-TCVAE. As can be seen, for the same celebA image, the reconstruction from the $\beta$-TCVAE is highly variable, illustrating the high variance of the learned posterior for $\beta$-TCVAE.

**??**

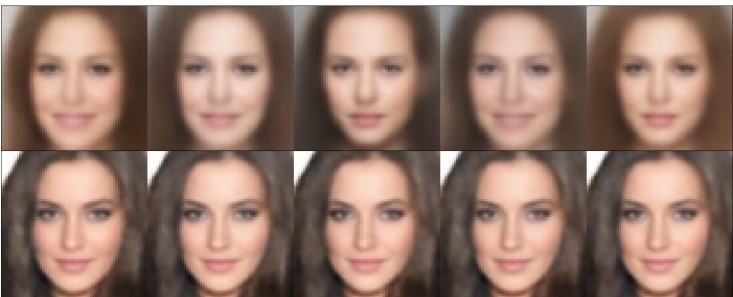

Figure 33: The top row is the same sample from the $\beta$-TCVAE and the bottom row is the corresponding reconstruction for MS-VAE. As can be seen in the figure, despite variability in the sampled Y's, MS-VAE is able to reconstruct the celebA images with little variance.

## J    PIXEL-WISE RECONSTRUCTION LOSS

For all the methods, we report the L1 and L2 reconstruction errors. As shown in Figures 34 and 35, pixel-wise reconstruction loss can be misleading. In the case of SmallNorb, where MS-VAE clearly has significantly better image fidelity (as determined by the reconstruction FIDs and traversal plots), the L1 and L2 error are only slightly better than TCVAE.

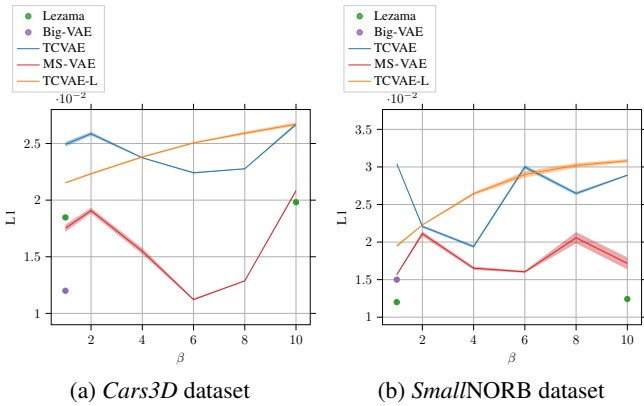

(a) *Cars3D* dataset          (b) *Small*NORB dataset

Figure 34: L1 (lower is better) comparison for $\beta$-TCVAE ($C \to Y$), $\beta$-TCVAE-L (latent dimensionality same as $C + Z$), and MS-VAE models.

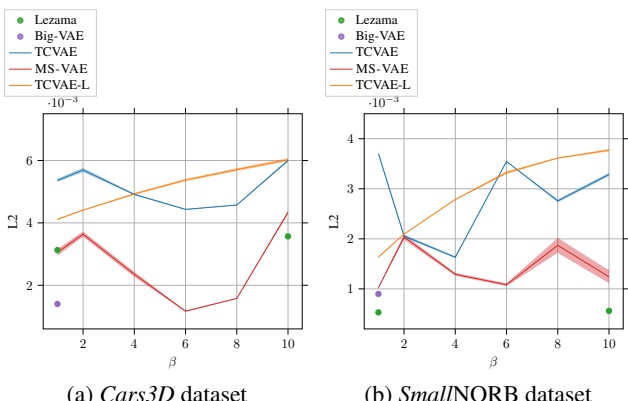

(a) *Cars3D* dataset          (b) *Small*NORB dataset

Figure 35: L2 (lower is better) comparison for $\beta$-TCVAE ($C \to Y$), $\beta$-TCVAE-L (latent dimensionality same as $C + Z$), and MS-VAE models.

## K    D-SEPARATION IN MS-VAE: AN ILLUSTRATIVE EXAMPLE

We provide a constructive example of the graphical model of MS-VAE shown in Figure 2c to illustrate the d-separation of $Y$ as discussed in Section 3.1.

Suppose we are given a dataset as shown in Figure 36d and we have trained a (disentanglement) model ($C \to Y$) where $C \sim \mathcal{U}$niform($[1, 2, 3]$) and $P(Y|C)$ is a deterministic mapping as shown in Figure 36a, where the indices above each image are the corresponding $C$s. As it can be seen, only a

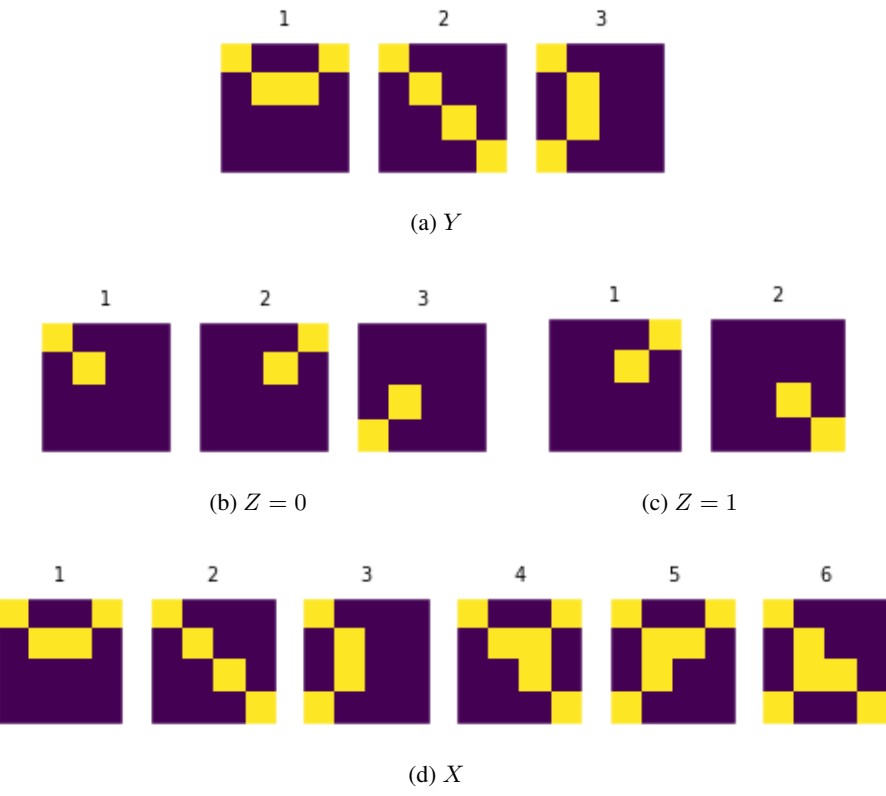

(a) $Y$

(b) $Z = 0$    (c) $Z = 1$

(d) $X$

Figure 36: A constructive example to illustrate d-separation in MS-VAE.

part of dataset (first three images) are modeled by the generative process of $C- > Y$. Now, with $Y$ generated, we add details to recover back the entire dataset. We first sample $Z \sim \mathcal{B}$ernoulli($0.5$) and add a mask to $Y$. The mask is *uniformly* sampled conditionally on $Z$ as in Figure 36b ($Z = 0$) and Figure 36c ($Z = 1$). To be more specific, e.g., given that we have sampled $Z = 0$, we apply a mask, uniformly chosen from Figure 36b to $Y$, which yield $X$. It's not hard to check that this generative process will yield a distribution with support shown in Figure 36d. Also, note that $X$ and $Y$ are in the same space. To simplify the discussion, we will represent the realizations of the random variable $X$ and $Y$ using the integer ($C$) above each image.

We want to check that $C$ and $Z$ are not conditionally independent given $X$ as implied by d-separation. To do so, we pick $X = 4$ and check $P(C, Z|X = 4) \overset{?}{=} P(C|X = 4)P(Z|X = 4)$. The probability distributions involved here can be represented by tables, as shown in Table 2. It can be verified that $P(C, Z|X = 4) \neq P(C|X = 4)P(Z|X = 4)$.

Now let's check the conditional independence if we *additionally* condition on $Y$, e.g. $Y = 1$. Specifically, we check $P(C, Z|X = 4, Y = 1) \overset{?}{=} P(C|X = 4, Y = 1)P(Z|X = 4, Y = 1)$. The probability distributions involved here can be represented by tables, as shown in Table 3. It can be verified that $P(C, Z|X = 4, Y = 1) = P(C|X = 4, Y = 1)P(Z|X = 4, Y = 1)$ holds. In other words, conditioning on $X = 4$ and $Y = 1$ d-seperates $C$ and $Z$.

| C | Z | Probability |
|---|---|---|
| 1 | 0 | 0 |
| 1 | 1 | 3/8 |
| 2 | 0 | 1/4 |
| 2 | 1 | 3/8 |
| 3 | 0 | 0 |
| 3 | 1 | 0 |

(a) $P(C, Z | X = 4)$

$\neq$

| C | Probability |
|---|---|
| 1 | 3/8 |
| 2 | 5/8 |
| 3 | 0 |

(b) $P(C | X = 4)$

$\times$

| Z | Probability |
|---|---|
| 0 | 1/4 |
| 1 | 3/4 |

(c) $P(Z | X = 4)$

Table 2: Conditioning on $X$ only.

| C | Z | Probability |
|---|---|---|
| 1 | 0 | 0 |
| 1 | 1 | 1 |
| 2 | 0 | 0 |
| 2 | 1 | 0 |
| 3 | 0 | 0 |
| 3 | 1 | 0 |

(a) $P(C, Z | X = 4, Y = 1)$

$=$

| C | Probability |
|---|---|
| 1 | 1 |
| 2 | 0 |
| 3 | 0 |

(b) $P(C | X = 4, Y = 1)$

$\times$

| Z | Probability |
|---|---|
| 0 | 0 |
| 1 | 1 |

(c) $P(Z | X = 4, Y = 1)$

Table 3: Conditioning on $X$ and $Y$.

## L    ABLATION STUDIES

We now evaluate four major modeling choices of MSVAE. Specifically, we performed ablation studies to understand the contribution of: 1) Our two-step training paradigm, 2) Our use of $Y$ instead of $C$ for the second stage, 3) Our use of AdaIN to incorporate $Z$, 4) Our decision to learn $C$ before learning $Z$. To evaluate our two-step training process, we compare MSVAE to a model where both the $C$ and $Z$ factors are learned together in an end-to-end manner (by penalizing only $C$) and $Z$ is incorporated using AdaIN. To evaluate how using $Y$ improves reconstruction performance over using $C$, we compare MSVAE to a version of MSVAE where $Y$ is replaced with $C$. To evaluate the contribution of AdaIN for disentanglement, we compare MSVAE to a version of MSVAE where $C$ and $Z$ are concatenated together as input to the second stage DGM. Finally, to evaluate our decision to learn $C$ before learning $Z$, we train a version of MSVAE where $Z$ is learned in the first stage and then $C$ is learned in the second stage ($C$ and $Z$ are concatenated together to reconstruct $X$). All experiments are performed on the *Cars3D* dataset.

### L.1    IMPORTANCE OF TWO-STEP TRAINING

MSVAE uses two-step training to ensure that the disentangled factors are captured in $C$ via the underlying $\beta$-VAE model and residual factors are captured in $Z$. We now show that the same will not happen if the latent space ($C$) of the underlying $\beta$-VAE is extended to incorporate $Z$ and then trained in an end-to-end fashion using the same AdaIN decoder. For this purpose we train a $\beta$-VAE model (BV) with the dimensionality of $C$ set to 5. As shown in Figure 37, BV discovers disentangled factors that capture azimuth, scale and elevation. Next, we train a model (M1) which has 10 latent factors. We enforce a higher $\beta$ penalty for 5 of these latent factors ($C$) and a normal VAE $\beta$ penalty for the other 5 factors ($Z$). The contribution of $Z$ is still limited using the AdaIN decoder strucutre. As can be seen in Figure 37, the disentangled factors $C$ in M1 are now entangled with other factors (e.g. identity, color). This experiment illustrates how end-to-end training cannot easily reproduce the two-step training scheme in MSVAE which induces d-separation of $C$ and $Z$.

### L.2    IMPACT OF USING $Y$ OVER $C$ AND ADAIN OVER CONCATENATION

MSVAE utilizes AdaIN to improve the reconstruction $Y$ from the underlying disentangled representation learner with information from the correlated factors $Z$. To understand the importance of using AdaIN (rather than concatenation) and $Y$ (instead of $C$), we perform the following experiment: Using a $\beta$-VAE model (BV) as the $C \rightarrow Y$ sub-graph, we train three models MSVAE-C (M2), MSVAE-C-IN (M3) and MSVAE (M4). In M2, we concatenate the inferred $C$ with $Z$ and use that as input to a DGM to reconstruct $X$. In M3, we still use the same $C$ but introduce $Z$ via AdaIN in the second stage DGM. In M4, we train MSVAE as done in the paper using $Y$ and AdaIN. In Figure 38, we show traversal plots of 4 of the disentangled factors for each of these models. For reconstruction quality, M4 outperforms both M3 and M2. This is expected as $Y$ is produced using both $C$ and the learned parameters of the first stage decoder $\theta$ and, therefore, contains much more semantic information about the observation $X$ than $C$ does. Using only $C$ requires the second stage DGM to relearn $\theta$ while also modelling the residual between $Y$ and $X$. For disentanglement, M4 and M3 both outperform M2. This illustrates how simply concatenating $Z$ and $C$ and inputting into into the second stage DGM will result in a non-linear entanglement of the two. This entanglement causes the network to fail to condition on $Y$ sufficiently and to use the entangled representation as a whole towards the reconstruction of $X$. Utilizing AdaIN to incorporate $Z$ (as done in M3 and M4), allows the network to use Z later in the generative process to model the residual information. This allows the network to better maintain the information in $Y$ (or $C$) while still incorporating $Z$.

### L.3    ORDERING OF $C$ AND $Z$

In MSVAE, we first learn the disentangled factors $C$ and then we learn the residual factors $Z$. In this ablation, we evaluate how the order in which these latent variables are learned affects the final disentanglement. For this purpose, we train MSVAE-ZC (M5) where, in the first stage, $Z$ is learned using a standard VAE. In the second stage DGM, the disentangled factors $C$ are learned by enforcing a KL-penalty and then concatenated with the learned $Z$ from the first stage to reconstruct $X$. In Figure 39, we show traversals of 4 of the disentangled factors $C$. As can be seen, this new model M5 does not capture any disentangled factors in $C$. In contrast, MSVAE (M4) and BV capture a

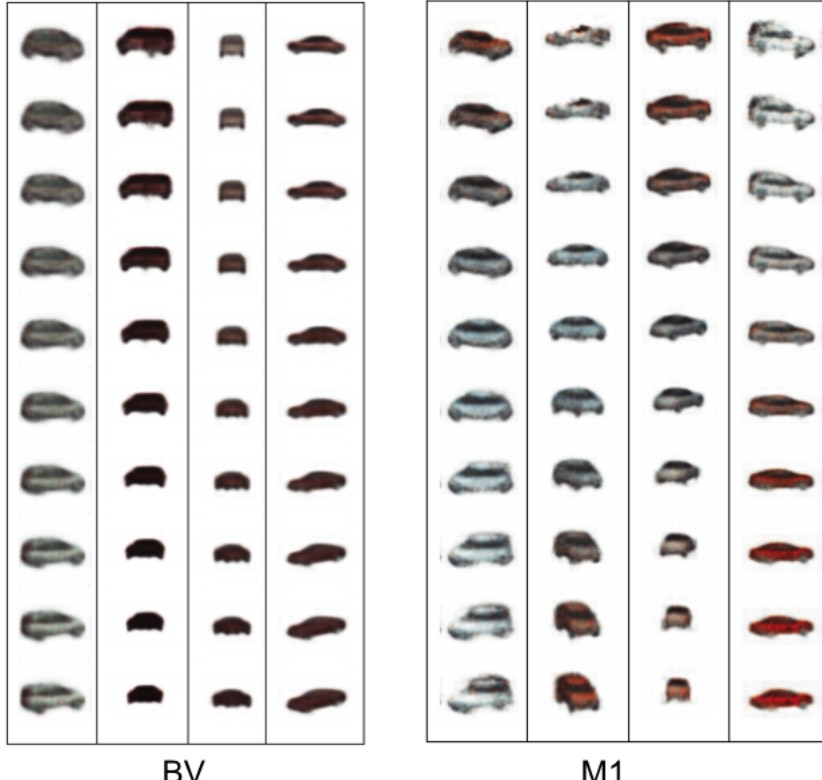

Figure 37: BV = $\beta$-VAE, M1 = $\beta$-VAE+Z. BV is trained with a dimensionality of $C = 5$. M1 is trained with a dimensionality of $C = 5$ and $Z = 5$. For M1, the $\beta$-penalty is only applied to $C$ and $Z$ is still incorporated using AdaIN. In this figure, we show traversals of 4 of the learned disentangled factors $C$ for both BV and M1. As can be seen, traversing $C$ for BV captures different disentangled factors (e.g. azimuth, scale, elevation). Traversing $C$ for M1, however, illustrates that the disentangled factors are now entangled with other factors (e.g. identity, color).

variety of disentangled factors in $C$ (e.g. Azimuth, scale, elevation). We hypothesize that M5 fails to extract any meaningful disentangled factors because $Z$ already captured both the entangled and disentangled factors in the first part of the training. This is clearly shown in Figure 40. As such, the second stage will suffer from the shortcut problem where the decoder utilizes $Z$ and does not learn anything meaningful in $C$ (due to the high KL-penalty).

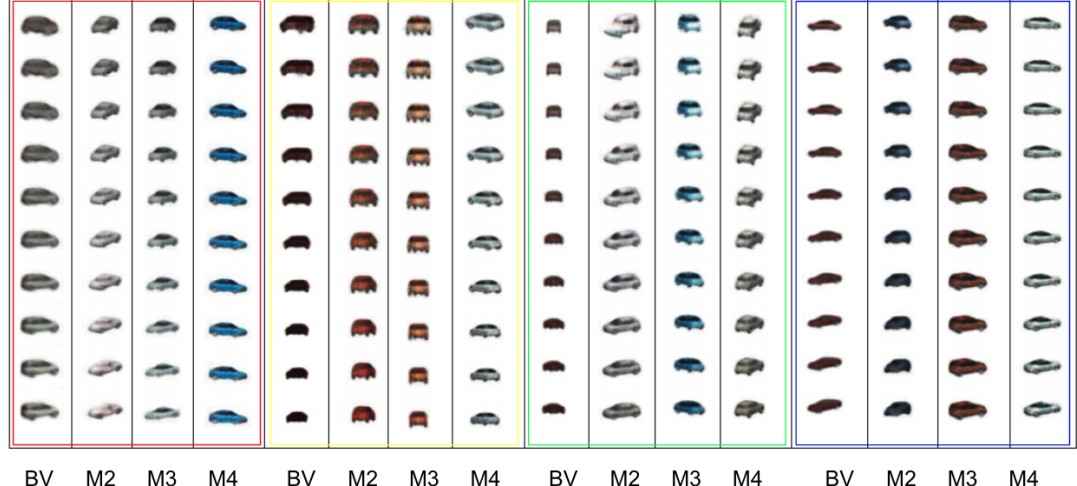

| BV | M2 | M3 | M4 | BV | M2 | M3 | M4 | BV | M2 | M3 | M4 | BV | M2 | M3 | M4 |

Figure 38: BV = $\beta$-VAE, M2 = MSVAE-C, M3 = MSVAE-C-IN, M4 = MSVAE. BV is trained with a dimensionality of $C = 5$. In this figure, we show traversals of 4 of the learned disentangled factors and how they are preserved in each version of MSVAE (each color box is one disentangled factor). To perform traversals for each MSVAE, we encode X using the $\beta$-VAE to get C and Y, then we extract Z using Y; We fix Z for the traversals of each MSVAE. Clearly, M4 and M3 preserve the disentanglement from BV better than M2 (M2 changes the identity and color). Also, M4 has the best reconstruction of the three MSVAE models as it utilizes Y to improve the reconstruction (rather than just using C). This ablation study illustrates that AdAIN is important for preserving disentanglement and using Y is important for improving reconstruction.

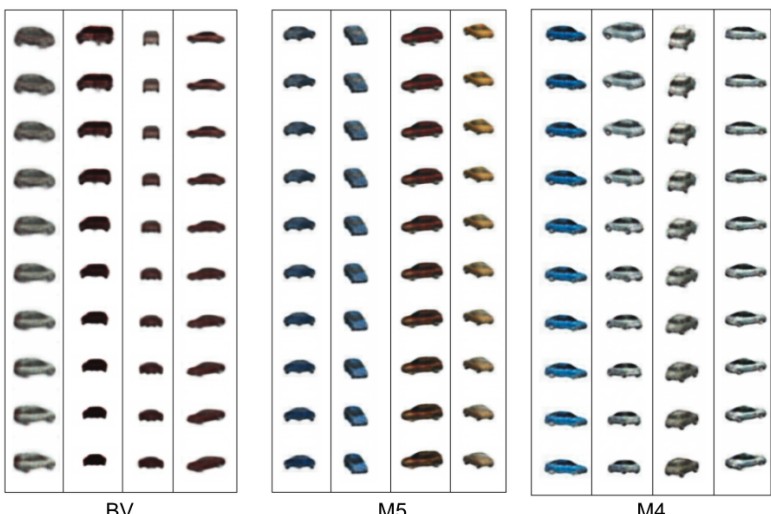

| BV | M5 | M4 |

Figure 39: BV = $\beta$-VAE, M5 = MSVAE-ZC, M4 = MSVAE. BV is trained with a dimensionality of $C = 5$. In this figure, we show traversals of 4 of the learned disentangled factors and how they are preserved in each of the three models. To perform traversals of MSVAE-ZC, we encode X using a VAE to first get Z and Y, then extract C using X. To perform traversals for MSVAE, we encode X using the $\beta$-VAE to get C and Y, then we extract Z using Y. We fix Z for the traversals in MSVAE. Clearly, M4 preserves the disentanglement from BV better than M5 (M5 as expected, is not disentangled).

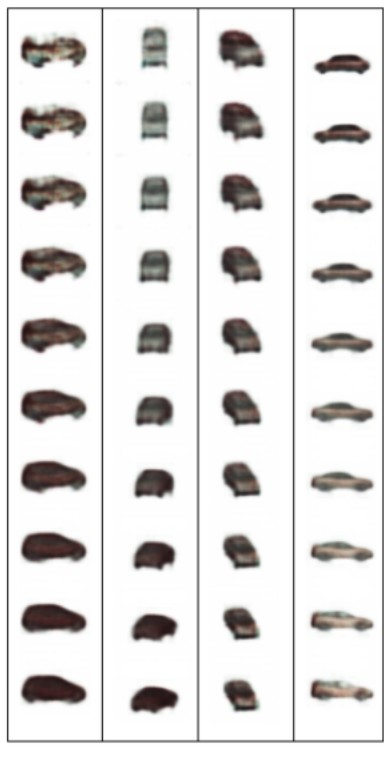

M5

Figure 40: M5 = MSVAE-ZC. In this figure, we show traversals of the $Z$ factor in MSVAE-ZC model. Clearly, $Z$ is capturing factors such as scale, identity, color, rotation in an entangled fashion.

