# OpenReview forum: "Improving the Reconstruction of Disentangled Representation Learners via Multi-Stage Modelling"
_ICLR.cc/2021/Conference — Reject_

### Official Review · AnonReviewer4 · 2020-10-26

**Rating:** 6
**Confidence:** 3

**Review:**

=======================================================================================

Summary :

Disentangled representation (DR) of data is useful in downstream tasks. However, VAE-based DR fundamentally suffers from a trade-off between high-quality reconstruction images and disentangling. To overcome this point, the paper approaches VAE from the multi-stage modeling (so-called MS-VAE). The proposed method starts from the other standard DR method which learns low-quality image(Y) and then, improves the quality of the image via training additional encoded representation(Z). The widely-used techniques in style transfer (FILM and AdaIN) are used to adopt 'Z' into 'Y' for a high-quality reconstructed image. The authors evaluated the proposed method through FID(high-quality image) and MIG(disentanglement). At the similar scale of complexity, the proposed method obtained high FID score and low MIG score than the baselines.

=======================================================================================

Reasons for score:

Overall, I vote for rejection. I'm not fully persuaded by the results from the novelty of this paper.
I left my concerns (see cons). Hopefully, the authors cover my concerns in the rebuttal period.

=======================================================================================

Strong points:

(1) Paper is well written.

(2) Authors provided the justification of the proposed method via a graphical model.

=======================================================================================

Cons :

The insight in the MS-VAE is quite overlapped in CascadeVAE [1]. CascadeVAE learns the continuous representation sequentially and shows that this sequential learning of representation helps disentanglement representation without special technique provided in FactorVAE, TC-\betaVAE, and etc. I suggest the authors provide comparison experiments with this kind of sequential learning in representation.

Also, an ablation study on the proposed method without AdaIN and FILM would increase the persuasion of readers and provide justification for the proposed method. At a high level, the overall method is to learn the representation of an original image which helps to transform a low-quality reconstructed image, which is trained in VAE, to a high-quality reconstructed image. Also, the experiment results in figure 4 show that disentanglement was not improved and only the quality of the image was improved.

Finally, if the ultimate motive of this paper is to learn the representation which can reconstruct the original image fairly well, the comparison with the VAE-based Method([2], [3])  which generated a high-quality image should be added.
For the disentangled representation which is used for future downstream tasks, there is no big difference with \beta-TC VAE in the best hyper parameter setting as shown in Figure 6.

[1] Learning discrete and continuous factors of data via alternating disentanglement, ICML2019.
[2] IntroVAE: Introspective Variational Autoencoders for Photographic Image Synthesis, NIPS2018.
[3] NVAE: A Deep Hierarchical Variational Autoencoder, NIPS 2020.

=========================================================================================================

After rebuttal (and a final question):


Thank you for your responses and the experiments I requested. I hope the authors would add discussions with M5 and quantitative results in the final version. Before I finalize my ratings, I have a final question after seeing supplementary material L. I believe M5 is enough to achieve the goal of this paper since $[Z, C]$ captures disentangled factors in $Z$ and $[Z, C]$ improves the reconstruction quality. Then, what is the advantage of MSVAE over M5?


=========================================================================================================

After the final question :

Thank you for your reply. I misunderstood the experiment M2 and propose M5. I believe M5 is just a variant of M2.
The authors should add quantitative results (FID and disentanglement score) comparing the baselines (in supplementary material L) in their final version. I would raise my score to 6.

Minor : M2-M4 denote mutual information (paper) and baselines (supp). Please use different notations.

---

> ### Author Response · Authors · 2020-11-15
> **Response**
>
> We thank R4 for the feedback and giving us a chance to address their concerns.
>
> > "The insight in the MS-VAE is quite overlapped in CascadeVAE [1]"
>
> We would like to clarify that CascadeVAE and MSVAE are very different methods, both in terms of approach as well as motivation.
> - CascadeVAE’s training method proposes to __alternate__ between learning the continuous disentangled representation Z and the discrete representation D. This alternating training scheme ensures that changes in the learned discrete representation D will change (improve) the learned continuous representation Z. As a result, CascadeVAE focuses on *improving disentanglement performance and NOT the reconstruction quality*.
> - MSVAE is a multi-stage generative model that __sequentially__ learns the disentangled representation C and then learns the residual representation Z. This sequential training approach ensures that learning the residual representation Z does not change the disentangled representation C. As a result, MSVAE focuses on *improving the reconstruction quality and NOT the disentanglement*.
>
> To further distinguish the two models, one could imagine training MSVAE with CascadeVAE as the first stage. This should lead to a model with equivalent disentanglement performance as CascadeVAE but with higher reconstruction quality.
>
> > "An ablation study on the proposed method without AdaIN and FILM would increase the persuasion of readers and provide justification for the proposed method."
>
> As noted in our response to Reviewer 2, when Z is concatenated to Y and inputted into the secondary network at the start of training, it will result in a non-linear entanglement between Y and Z. Once entangled this way, the network will fail to condition on Y sufficiently and use the entangled representation as a whole towards the reconstruction of X. To overcome this, we induced an architectural bias with our FiLM/AdaIN architecture. Specifically, by inputting only Y into the decoder and incorporating Z through AdaIN at each layer, the network is able to use Z later in the generative process to model the residual information (as explained in [1]). This allows the network to better utilize the information in Y. We have amended our explanation in Section 3.2 to better explain this phenomenon.
>
> [1] Dumoulin, Vincent, et al. "Feature-wise transformations." Distill 3.7 (2018): e11.
>
> > “The experiment results in figure 4 show that disentanglement was not improved and only the quality of the image was improved.”
>
> There seems to be a misunderstanding of the motivation of our work. The explicit goal of MSVAE is to improve the reconstruction of preexisting disentangled representation learning methods without hurting disentanglement performance. Figure 4 demonstrates that MSVAE achieves this stated goal by improving the image quality while maintaining the disentanglement.
>
> > "For the disentangled representation which is used for future downstream tasks, there is no big difference with $\beta$-TC VAE in the best hyper parameter setting as shown in Figure 6."
>
> There seems to be a misunderstanding of the results shown in Figure 6. In Figure 6, we display the accuracy of three shallow MLPs that are trained on the disentangled factors C, the correlated factors Z, and the concatenation of the two factors C+Z, to predict object properties of SmallNORB (e.g. category, instance, elevation, etc.). Importantly, the MLP with the highest accuracy is trained on the combined representation C+Z, highlighting that Z encodes semantically useful information which is complementary to the information learned in C. Therefore, utilizing both C and Z from MSVAE would potentially *improve* performance on downstream tasks rather than just using C or Z separately.
>
> > "The comparison with the VAE-based Method([2], [3]) which generated a high-quality image should be added."
>
> We do not think that comparing to these methods would provide any additional insight into our method. This is because the main motivation of MSVAE is to improve the reconstruction of disentangled representation learning methods *without hurting the disentanglement performance*. IntroVAE and NVAE are exclusively designed to produce high-quality reconstructions, but have no ability to learn disentangled representations. Also, please note that MSVAE is a general method that can be implemented with any combination of disentangled representation learning method (C → Y) and upsampling generative model ((Y,Z) → X). Therefore, one could even use NVAE as the second stage of MSVAE to further improve reconstruction. The generality of our method is highlighted in the appendix where we show how using GANs and FLOW-based models for the second stage of MSVAE can lead to even higher quality reconstructions.

---

> > ### Comment · AnonReviewer4 · 2020-11-23
> > **Response to response**
> >
> > Thank you for the response which helps my understandings of this paper. However, I believe the authors misunderstood some of my concerns. Therefore, I need to clarify my concerns again and I hope the authors respond to my concerns.
> >
> > > The insight in the MS-VAE is quite overlapped in CascadeVAE [1]
> >
> > First, in [1], there are two main ideas :
> > *  Learning the continuous representation sequentially
> > *  Learning the discrete and continuous representation alternatively.
> >
> > I want to address the first idea which learns the continuous representation sequentially.
> >
> > > An ablation study on the proposed method without AdaIN and FILM would increase the persuasion of readers and provide justification for the proposed method.
> >
> > I have read supplementary material L. However, I suggest the different baseline experiments which help the understanding of future readers or other reviewers.
> >
> > First, assume that there are two encoders ($\text{Enc}_1$ and $\text{Enc}_2$) and two decoders ($\text{Dec}_1$ and $\text{Dec}_2$).
> >
> > Stage1 : Train $\text{Enc}_1$ and $\text{Dec}_1$ (This stage is the same in the paper).
> >
> > * Encoding : $\text{Enc}_1$ : $X \to Z$.
> > * Decoding : $\text{Dec}_1$ : $Z \to Y$.
> >
> > Stage2 : Fix $\text{Enc}_1$ and $\text{Dec}_1$ and Train $\text{Enc}_2$ and $\text{Dec}_2$.
> >
> > * Encoding : $\text{Enc}_2$ : $X \to C$.
> > * Decoding : $\text{Dec}_2$ : $[Z,C] \to Y'$ ($Z=\text{Enc}_1(X)$).
> >
> > Use $Z,C$ as the disentangled representation of $X$ which can reconstruct $X$ with high quality. I call this baseline $M5$.
> >
> > I believe the performance of $M5$ and $MSVAE$ should be evaluated with respect to disentanglement score (disentanglement measure) and FID (quality measure) as in Figure3.
> >
> > To this end, I believe this baseline introduced above ($M5$) is crucial to determine the acceptance of this paper.

---

> > > ### Author Response · Authors · 2020-11-23
> > > **Response**
> > >
> > > Once again, we thank R4 for their feedback and for engaging with us to help improve our paper.
> > >
> > > > I want to address the first idea which learns the continuous representation sequentially.
> > >
> > > The idea of “learning continuous representation sequentially” refers to two different procedures in CascadeVAE and MSVAE.
> > >
> > > In CascadeVAE, the high-$\beta$ penalty for each latent dimension is annealed in a sequential manner. They claim that is an improvement over a similar disentangled representation learning method ([5]) that also anneals the $\beta$-penalty but not sequentially. However, they show that [4], which does not use sequential annealing, performs competitively to CascadeVAE-C (table 1 in the paper). As shown in prior work [5,6], reducing $\beta$ during the course of training helps improve the reconstruction quality by encoding a bit more of the correlated factors. But this does not alleviate the trade-off between disentanglement and reconstruction quality as both correlated and disentangled factors are forced to be encoded together and there is no theoretical reason for the posterior to factorise. This is exactly the problem that we solve in MSVAE using a multi-step approach.
> > >
> > > In MSVAE, sequential learning refers to the fact that the model first learns a set of disentangled factors $C$ and then uses another DGM to model the residual correlated factors $Z$. This multi-step training procedure ensures that there is no trade-off between disentanglement and reconstruction quality. This is because of the d-separated graphical model of MSVAE which, unlike CascadeVAE and other annealing based approaches, provides a theoretical justification for separating the learning of disentangled and correlated factors.
> > >
> > > We have added a citation for CascadeVAE in the prior work section and can elaborate on the differences between the two models if R4 thinks that will be valuable to the reader.
> > >
> > > [4]. Gao, S., Brekelmans, R., Ver Steeg, G., & Galstyan, A. (2019, April). Auto-encoding total correlation explanation. In The 22nd International Conference on Artificial Intelligence and Statistics (pp. 1157-1166). PMLR.
> > >
> > > [5] Burgess, Christopher P., et al. "Understanding disentangling in $\beta $-VAE." arXiv preprint arXiv:1804.03599 (2018).
> > >
> > > [6] Shao, Huajie, et al. "Controlvae: Controllable variational autoencoder." Proceedings of the 37th International Conference on Machine Learning (ICML). 2020.
> > >
> > > > I suggest the different baseline experiments which help the understanding of future readers or other reviewers.
> > >
> > > As recommended by R4, we have added an ablation study to Appendix L that evaluates our decision to learn $C$ before learning $Z$. For this purpose we trained the proposed model M5 where we first learn $Z$ using a vanilla VAE (Encoder 1 and Decoder 1). Then, fixing Encoder 1, we train Encoder 2 to learn the disentangled factors $C$ (by enforcing a KL-penalty) and to reconstruct $X$ by inputting the concatenation of $C$ and $Z$ into Decoder 2. As shown in Figure 39, M5 fails to learn any disentangled factors. This is because the unconstrained continuous factors $Z$ already explain both disentangled and correlated factors of generation (shown in Figure 40). As such, the second stage will suffer from the shortcut problem where the decoder utilizes $Z$ and does not learn anything meaningful in $C$ (due to the high KL-penalty).
> > >
> > > Given that the qualitative results very clearly show the loss of disentanglement for M5, in the interest of time, we did not add any quantitative evaluations. However, if R4 would still like us to add the quantitative scores after seeing the qualitative results, we would be happy to oblige.

---

> > > > ### Comment · AnonReviewer4 · 2020-11-25
> > > > **Final question**
> > > >
> > > > Thank you for the response.
> > > >
> > > > I have read supplementary material L and the results raised my final question.
> > > > I believe M5 is enough to achieve the goal of this paper since  $[Z, C]$ captures disentangled factors in $Z$ and $[Z, C]$ improves the reconstruction quality. Then, what is the advantage of MSVAE over M5?

---

> > > > > ### Author Response · Authors · 2020-11-25
> > > > > **Response**
> > > > >
> > > > > >  I believe M5 is enough to achieve the goal of this paper since [Z,C] captures disentangled factors in Z and [Z,C] improves the reconstruction quality. Then, what is the advantage of MSVAE over M5?
> > > > >
> > > > > Just to clarify, the first stage of M5 is a standard VAE (i.e $\beta=1$) where $Z$ is learned. This results in $Z$ learning an entangled representation of $X$, i.e. traversal of $Z$ will change more than one factors of variations in the generation, as shown in Figure 40. Notice how in all the columns changing one factor at a time is changing not just the orientation (azimuth, elevation, scale) but also car type and color. For example, in the last column of M5 in Figure 40, traversing the last factor in $Z$ randomly changes the type of the car as well as rotation.
> > > > >
> > > > > Then in the second stage of M5, we extract $C$ using a different encoder with $\beta=4$. $Z$ and $C$ are concatenated together to decode $X$. As shown in Figure 39, $C$ does not learn any disentangled information about $X$ as well, due to the shortcut problem (the decoder uses $Z$ and ignores $C$). __Therefore, neither $C$ nor $Z$ in M5 learn disentangled factors__. Clearly, MSVAE provides a benefit over M5 because $C$ does encode disentangled factors in MSVAE.
> > > > >
> > > > > In case the M5 model we described is not what you had in mind and you wanted us to learn disentangled factors in $Z$ by using a higher $\beta$, we have already run this experiment. Please see L.2 for this experiment. This model (M2) also does not perform as well as MSVAE.

---

> ### Author Response · Authors · 2020-11-25
> **Final Response**
>
> Thank you very much for the feedback and for increasing your score. We will add the quantitative results for the ablation study in the final version and will update the notation for the models in the appendix.

---

### Official Review · AnonReviewer1 · 2020-10-26
**Good paper, but lacks some details on the motivation of the proposed approach**

**Rating:** 6
**Confidence:** 4

**Review:**

This paper proposes a multi-stage approach to learning disentangled representation that refines the low quality reconstructions of a state of the art architecture using a second deep generative model that learns how to modify these reconstructions with additional correlated latent variables.

The paper is well written and easy to follow. The introduced idea is quite straightforward, I would not be surprised at all if it was introduced before, but I do not recall seeing it in any other work so I will regard it as novel (unless the other reviewers point out similar works). Despite its simplicity, this method makes a noticeable difference in the experiments, so it could be very useful trick for other researchers working in the area of learning disentangled representations.

My main concern for this paper is the motivation of the proposed approach. The need for a multi-stage approach is justified using d-separation, saying that in this way independent and correlated latent variables (c and z) are completely separated during training. This is true, but is the multi-stage approach the best/only way to obtain this effect?
It would be useful to understand whether a similar effect could be obtained instead in an end to end fashion with the right architecture or training procedure, since I believe that such a procedure would have a much bigger impact for other researchers.
An interesting ablation study that would allow to better understand the mechanics of this technique would be for example using the same split of c and z and the same FiLM + AdaIN decoder, but training end to end placing the beta factor as in the beta-VAE only on the c latent variables, with for example a schedule for beta in which independent latent variables are more important at the beginning of training, or some alternating updates.

---

> ### Author Response · Authors · 2020-11-15
> **Response**
>
> We thank R1 for their feedback.
> > “... is the multi-stage approach the best/only way to obtain this effect?... An interesting ablation study that would allow to better understand the mechanics of this technique would be for example using the same split of c and z and the same FiLM + AdaIN decoder, but training end to end placing the beta factor as in the beta-VAE only on the c latent variables...”
>
> While an end-to-end approach is almost always preferable to training the components separately, unsupervised end-to-end training is not possible with our proposed graphical model (Figure 2c) *without knowing Y apriori*.  This is because Y must be observed to d-separate the latent variables C and Z, preventing any disruptive interference. Without knowing Y apriori, the reconstruction from the first stage of our model acts as this observed variable, separating the latent spaces of the two models. However, when both stages are trained together, there is no theoretical guarantee that C and Z will be disentangled even if you enforce some posterior penalty. The proposed ablation study i.e. learning a map (Y,Z) → X without fixing Y is, in general, very hard [1, JointVAE vs AAE-S]. In our particular case, the objective of this map is to regress Y to X using the residual information in Z. Clearly, this regression is not well defined if Y keeps changing during the training. As this question was asked by multiple reviewers, we have added an explanation to Section 3.1.
>
> [1] Learning discrete and continuous factors of data via alternating disentanglement, ICML2019.

---

> > ### Author Response · Authors · 2020-11-23
> > **Response: Ablation Study**
> >
> > For some reason, we could not comment directly to the reviewer's comment below. So we are posting our response to R1's response from 21 Nov here.
> >
> > We have now added an ablation study which illustrates how end-to-end training cannot easily reproduce the two-step training scheme in MSVAE which induces d-separation of $C$ and $Z$. As can be seen in Figure 37, the disentangled factors $C$ become entangled with other factors (e.g. identity, color). This new experiment has been added to Appendix L.
> >
> > If R1 could elaborate on any remaining concerns or questions they have with the paper, we are happy to address them during this rebuttal period.

---

### Official Review · AnonReviewer2 · 2020-10-28
**An interesting article and decent contribution, but a few points need clarifying**

**Rating:** 6
**Confidence:** 3

**Review:**

This article introduces a method for learning high-quality generative model with disentangled latent representation by splitting the learning process into two steps. The first step consists in learning a generative model on the data using a method with strong disentanglement constraints, producing a low-quality generation. As a second step, a conditional generative model is trained to turn this low-quality sample into an high quality one. This intermediate generation acts as an observed variable, and thus separates the latent spaces of the two models, effectively preventing disruptive interference in the learning of the "independent factors" on the one hand and the "dependent factors" on the other, as has been previously observed as a difficulty in the literature. The authors provide detailed empirical analysis of the performance of the model.

Overall I would say that this is an interesting article, and a decent contribution to the conference. The design of the model is explained in detail, and its position among the literature is explained in detail. The method in itself is pretty general, and the authors provide explanation on how to apply it in non-VAE contexts (Flows & GANs mostly) in appendix.

However, I have a few questions, and I think these points should be clarified in the text of the article:

1. The use of FiLM structure of the conditional VAE is motivated by the risk that the model would ignore Y and solely rely on Z to generate X. However, this seems rather counter-intuitive from a theoretical point of view: the training objective of the VAE rewards storing as little information as possible in the latent space. If the model can reliably use information present in Y, we should thus expect it to do so. The previously observed issue of models ignoring their disentangled latent and storing everything in the correlated one does not applies as, this is the point of your model, Y is here treated as if it was part of the dataset. So, is this ignoring of Y something you observed experimentally before deciding to use FiLM structures?

2. As a follow-up of the previous point, it would seem natural to expect that an abstract value for Y would potentially be more easily used by the conditional model than a low-quality reconstruction. Going further with that, one could consider using directly C as input to the conditional model: training β-TCVAE as you describe, and only keeping its encoder to produce a dataset of (X, C), then using this dataset to train the second stage as a conditional generative model p(X | C, Z) (with an encoder q(Z | X, C)). As per the previous argument, this second stage should be using C as much as possible. How do you think that would compare to MS-VAE?

3. Regarding the simple pendulum example cited in the text, it is unclear how Y0 and Y1 are produced. Given how precisely they are defined in the appendix, I'm inferring that they are ground truth data produced by the simulator. If that is the case, it seems to me that the structure presented here is no different from a regular latent variable model with multiple observed variables, as the main defining component of your approach (learning Y with a disentanglement-inducing method) is not present at all. If on the other hand Y0 and Y1 are indeed directly learned from θ, then it should be clarified.

-----------------

Minor remarks and typos:

- the legend for figure (2d) is mislabeled (c)
- the legend for figure (3e) seems to invoke the wrong dataset (the images look like from Cars3D, not SmallNORB)
- bottom of page 7, the text invokes figure (3h), which does not exist

---

> ### Author Response · Authors · 2020-11-15
> **Response**
>
> We thank R2 for the feedback and kind words. We have updated the minor mistakes/typos.
> > “The use of FiLM structure of the conditional VAE is motivated by the risk that the model would ignore Y and solely rely on Z to generate X. This seems rather counter-intuitive from a theoretical point of view...”
>
>
> This is a great point. If the secondary network in MSVAE could use the information in Y apriori, the VAE objective will ensure that Z only models the residual. However, when Z is concatenated to Y and inputted into the network at the start of training, it will result in a non-linear entanglement between Y and Z. Once entangled this way, the network will fail to condition on Y sufficiently and use the entangled representation as a whole towards the reconstruction of X. To overcome this, we induced an architectural bias with our FiLM/AdaIN architecture. Specifically, by inputting only Y into the decoder and incorporating Z through AdaIN at each layer, the network is able to use Z later in the generative process to model the residual information (as explained in [1] and often used in Style-Transfer). This allows the network to better utilize the information in Y. We have amended our explanation in Section 3.2 to better explain this phenomenon.
>
> [1] Dumoulin, Vincent, et al. "Feature-wise transformations." Distill 3.7 (2018): e11.
>
> > “... one could consider using directly C as input to the conditional model: training β-TCVAE as you describe, and only keeping its encoder to produce a dataset of (X, C), then using this dataset to train the second stage as a conditional generative model p(X | C, Z) (with an encoder q(Z | X, C))”
>
> This is a very good point and something we tried out early on. While we agree that C could be used as input to the conditional model (rather than Y), we decided to use Y since it is produced using both C and the learned parameters of the first stage decoder $\theta$ and, therefore, contains much more semantic information about the observation X than C does. Therefore, if one only uses C, the conditional model would have to simultaneously relearn $\theta$ while also modelling the residual between Y and X; this is a much harder task. Using Y allows the conditional model to focus on modelling the residual between X and Y.
>
>
> > “... it is unclear how Y0 and Y1 are produced… it seems to me that the structure presented here is no different from a regular latent variable model”
>
> In our pendulum example, the observed variables Y0 and Y1 are produced through supervised training of two networks. The first network is trained to produce the undamped angular displacement (Y0) given the pendulum length (L). The second network is trained to produce the angular displacement of a damped pendulum with a fixed mass (Y1) given the undamped angular displacement (Y0) from the previous network and the damping coefficient (b). *The combination of these two networks can be thought of as a supervised disentanglement method and as the first stage of MSVAE*. Then, for the second stage of MSVAE, we use an auxiliary DGM (with a latent variable Z) to model the residual between the reconstruction from this first stage and the output of a fully realistic pendulum simulation (which includes the pendulum length, damping coefficient, AND the mass). We show that the learned latent space Z is correlated with the mass and disentangled from the pendulum length and damping coefficient (please see our latent traversals in Figure 15 in the appendix).

---

> > ### Comment · AnonReviewer2 · 2020-11-20
> > **Thanks for your clarifications**
> >
> > I don't have any further questions, and I maintain my assessment.

---

> > > ### Author Response · Authors · 2020-11-23
> > > **Response: Ablation Studies**
> > >
> > > We have now added an ablation study to evaluate how using $Y$ improves reconstruction performance over using $C$ and how using AdaIN (instead of concatenation) leads to improved disentanglement. This new experiment has been added to Appendix L.
> > >
> > > If R2 has any remaining concerns about the paper/ablation studies, we are happy to address them during this rebuttal period.

---

### Official Review · AnonReviewer3 · 2020-11-02

**Rating:** 6
**Confidence:** 3

**Review:**

The paper studies the problem of learning disentangled representations while maintaining good data reconstruction. As common modeling, the latent representation is decomposed into disentangled representation C and correlated representation Z. Then a hierachical generative process is proposed, where the first stage is to reconstruct a preliminary version of the data given the disentangled representation C, and the second step is to reconstruct a full version of the data given C and correlated representation Z. The two stages are learned separately, with the first stage using the previous β-TCVAE model to learn C, and the second stage using the Feature-wise Linear Modulation (FiLM) technique.

Experiments show the approach can learn disentangled representation C as well as the previous β-TCVAE model, but improves in terms of the reconstruction quality. It is also shown that the second stage can properly maintain the conditioning on C (instead of conditioning only on Z).

As above, overall, the proposed modeling is simple and reasonably sound, and the experimental results / analyses are interesting and promising.

A potential shortcoming is that the two stage components are learned separately instead of jointly, which could results in suboptimial learning.

---

> ### Author Response · Authors · 2020-11-15
> **Response**
>
>
> We thank R3 for their feedback.
> > “... is the multi-stage approach the best/only way to obtain this effect?....An interesting ablation study...”
>
> While an end-to-end approach is almost always preferable to training the components separately, unsupervised end-to-end training is not possible with our proposed graphical model (Figure 2c) *without knowing Y apriori*.  This is because Y must be observed to d-separate the latent variables C and Z, preventing any disruptive interference. Without knowing Y apriori, the reconstruction from the first stage of our model acts as this observed variable, separating the latent spaces of the two models. However, when both stages are trained together, there is no theoretical guarantee that C and Z will be disentangled even if you enforce some posterior penalty. The proposed ablation study i.e. learning a map (Y,Z) → X without fixing Y is, in general, very hard [1, JointVAE vs AAE-S]. In our particular case, the objective of this map is to regress Y to X using the residual information in Z. Clearly, this regression is not well defined if Y keeps changing during the training. As this question was asked by multiple reviewers, we have added an explanation to Section 3.1.
>
> [1] Learning discrete and continuous factors of data via alternating disentanglement, ICML2019.
>
> ### Update
> We have now added an ablation study which illustrates how end-to-end training cannot easily reproduce the two-step training scheme in MSVAE which induces d-separation of $C$ and $Z$. As can be seen in Figure 37, the disentangled factors $C$ become entangled with other factors (e.g. identity, color). This new experiment has been added to Appendix L.

---

### Author Response · Authors · 2020-11-23
**Ablation Studies Added**

We have now added all the ablation studies for which the reviewers asked. Please refer to the appendix L for these new experiments. Specifically, we performed ablation studies to understand the contribution of: 1. Our two-step training paradigm (requested by R1 and R3), 2. Our use of $Y$ instead of $C$ for the second stage (requested by  R2), 3. Our use of AdaIN to incorporate $Z$ (requested by R2). To evaluate our two-step training process, we compare MSVAE to a model where both the $C$ and $Z$ factors are learned together in an end-to-end manner (by penalizing only $C$) and $Z$ is incorporated using AdaIN. To evaluate how using $Y$ improves reconstruction performance over using $C$, we compare MSVAE to a version of MSVAE where $Y$ is replaced with $C$. Finally, to evaluate the contribution of AdaIN for disentanglement, we compare MSVAE to a version of MSVAE where $C$ and $Z$ are concatenated together as input to the second stage DGM. All experiments are performed on the *Cars3D* dataset. We hope that adding these extra experiments will help future readers better understand the modelling and implementation choices that were made.

### Update:

In response to R4’s suggestions, we have added an ablation study to evaluate our decision to learn $C$ before learning $Z$. For this purpose, we train a version of MSVAE where $Z$ is learned in the first stage and then $C$ is learned in the second stage ($C$ and $Z$ are concatenated together to reconstruct $X$). We added this experiment to Appendix L.

---

### Decision · Program_Chairs · 2021-01-07
**Final Decision**

**Decision:**

Reject

**Comment:**

This paper presents a two-step approach to achieve disentangled representation and good reconstruction at the same time in deep generative models: the first step focuses on good disentanglement (e.g., with beta-TCVAE) while possibly sacrificing reconstruction reconstruction, while the second step focuses on high-quality reconstruction, conditioned on the low-quality reconstruction from disentangled representation. In this paper, each step uses an existing method: beta-TCVAE is used for the first step and AdaIN is used for the second, so the paper presents an intuitive combination of two existing methods to achieve both goals. Some useful ablation studies are provided to empirically justify the specific method choices. The concern is whether the two step approach is necessary to achieve both; the authors' argument is that models learning only one set of latent variables may not have the capacity to achieve both goals, and methods jointly learning two set of variables ("disentangled" and "correlated" variables) can not guarantee they represent disjoint structures of data. Some of these statements seem somewhat handwavy (including the d-separation argument, I am not very sure if it applies when the variables are learned separately), and shall be made rigorous and justified (theoretically and/or empirically).

The reviewers rate this paper to be borderline.